# Fine-Grained Dynamic Framework for Bias-Variance Joint Optimization on Data Missing Not at Random

**Mingming Ha**
MYbank, Ant Group
Beijing, China
`hamingming.hmm@mybank.cn`

**Xuewen Tao**
MYbank, Ant Group
Shanghai, China
`xuewen.txw@mybank.cn`

**Wenfang Lin**
MYbank, Ant Group
Hangzhou, China
`moxi.lwf@mybank.cn`

**Qiongxu Ma**
MYbank, Ant Group
Shanghai, China
`qiongxu.mqx@mybank.cn`

**Wujiang Xu**
MYbank, Ant Group
Shanghai, China
`xuwujiang.xwj@mybank.cn`

**Linxun Chen**
MYbank, Ant Group
Beijing, China
`linxun.clx@mybank.cn`

## Abstract

In most practical applications such as recommendation systems, display advertising, and so forth, the collected data often contains missing values and those missing values are generally missing-not-at-random, which deteriorates the prediction performance of models. Some existing estimators and regularizers attempt to achieve unbiased estimation to improve the predictive performance. However, variances and generalization bound of these methods are generally unbounded when the propensity scores tend to zero, compromising their stability and robustness. In this paper, we first theoretically reveal that limitations of regularization techniques. Besides, we further illustrate that, for more general estimators, unbiasedness will inevitably lead to unbounded variance. These general laws inspire us that the estimator designs is not merely about eliminating bias, reducing variance, or simply achieve a bias-variance trade-off. Instead, it involves a quantitative joint optimization of bias and variance. Then, we develop a systematic fine-grained dynamic learning framework to jointly optimize bias and variance, which adaptively selects an appropriate estimator for each user-item pair according to the predefined objective function. With this operation, the generalization bounds and variances of models are reduced and bounded with theoretical guarantees. Extensive experiments are conducted to verify the theoretical results and the effectiveness of the proposed dynamic learning framework.

## 1 Introduction

In virtually all real-world applications, the pieces of data we collected are partially missing with certain probabilities. A special case with the identical missing probability is known as missing at random (MAR) [1]. However, in online recommendation, search, and display advertising, there are lots of missing-not-at-random (MNAR) click, conversion, and rating records [2, 3, 4], which are missing with different probabilities, i.e., propensities. For example, in recommendation systems, a user usually clicks the items that she/he is likely to purchase and ignores other items with a low willingness to buy. Therefore, the observed click and conversion data is MNAR, which are not representative samples of all the events [5]. When the MNAR data is used to train a model, the prediction performance of this model on the MAR data is generally unacceptable. This is because MNAR data introduces sample selection bias [3, 6] into the prediction model. To eliminate sample selection bias, lots of debiasing estimators [3, 6, 7, 8, 9] have been developed, e.g., Error-Imputation-

Based (EIB) approach [10], Inverse Propensity-Scoring (IPS) technique [6], Doubly Robust (DR) method [11], and so forth.

However, in almost all debiased methods, the existence of propensities results in the high variance and generalization bound. [11, 12]. Therefore, various methods [5, 8, 12] have been developed to reduce estimation variances and improve the model stability. Even so, they still suffer from unbounded variances and generalization bounds when the propensity tends to zero. For the high variance and generalization bound caused by small propensities, some approaches compromise to self-normalized technique [12, 13] at the expense of unbiasedness. In addition, the overwhelming majority of previous works focus on the specific designs of the estimators or regularizers to reduce variance or eliminate bias while neglecting both the bias-variance relationship of estimators and the essence of the estimator designs.

In this paper, we reveal limitations of general regularization techniques. We find that it is impossible to reduce variance without sacrificing unbiasedness by introducing regularizers, and that regularization cannot guarantee estimators to have bounded variance and generalization bound. Besides, for general estimators, unbiasedness will inevitably result in unbounded variance and generalization bound. To some extent, the generalization bound can reflect the predictive performance of an estimator. Therefore, reducing and bounding the generalization bound can assist in improving the predictive performance of models. Since the generalization bounds of estimators contain the bias and variance terms, the essence of estimator design is not merely about eliminating bias, reducing variance, or simply achieving a bias-variance trade-off but about the quantitative joint optimization of bias and variance. Then, we develop a systematic dynamic learning framework to achieve this objective. To the best of our knowledge, this is the first work to systematically reveal limitations of general regularizers and the design perspective of the quantitative bias-variance joint optimization. Our main contributions can be summarized as follows:

1) We theoretically elaborate limitations of regularization techniques, and the relationship of unbiasedness, variance and generalization bound of general estimators.

2) Based on the general laws, we elaborate a novel design perspective for the estimator, namely the quantitative bias-variance joint optimization;

3) We develop a comprehensive dynamic learning framework with the bounded variances and generalization error to optimize a weighted objective with respect to bias and variance for each user-item pair $(u, i)$, which dynamically selects different estimators for different user-item pair from a family of estimators according to the given objective function;

4) We conduct extensive experiments to verify the theoretical results and the performance of the dynamic regularizer and estimators.

## 2 Preliminaries

**Data missing not at random.** Denote the sets of users and items as $\mathcal{U} = \{u_1, u_2, \ldots, u_M\}$ and $\mathcal{I} = \{i_1, i_2, \ldots, i_N\}$, respectively. The set of all user-item pairs is denoted as $\mathcal{D} = \mathcal{U} \times \mathcal{I}$. Define the true and prediction matrices as $Y \in \mathbb{R}^{M \times N}$ and $\hat{Y} \in \mathbb{R}^{M \times N}$, where prediction tasks include rating, CTR and CVR predictions, and so forth. Each element $y_{u,i}$ in $Y$ and each entry $\hat{y}_{u,i}$ in $\hat{Y}$ are the true label and predicted output of a user $u$ to an item $i$. In general, it is impossible to observe all entries in the matrix $Y$. The indicator entry of revealed elements is defined as $o_{u,i} \in \{0, 1\}$. If the true label $y_{u,i}$ is revealed, the indicator entry of $(u, i)$ satisfies $o_{u,i} = 1$. If an entry in $Y$ is missing, then $o_{u,i} = 0$. The corresponding indicator set is denoted as $\mathcal{O} = \{o_{u,i} = 1\}$. Considering the case that no entries are missing, the prediction inaccuracy [11] of $\hat{Y}$ is defined as

$$L_{\text{real}}(\hat{Y}, Y) = \frac{1}{MN} \sum_{u=1}^{M} \sum_{i=1}^{N} e_{u,i} = \frac{1}{|\mathcal{D}|} \sum_{(u,i) \in \mathcal{D}} e_{u,i},$$

where $e_{u,i}$ is the prediction error. $e_{u,i}$ can be selected as mean absolute error (MAE), mean square error (MSE) or other measures. The objective of prediction problems is to minimize the prediction inaccuracy $L_{\text{real}}(\hat{Y}, Y)$ [5, 7, 8, 11, 12, 14]. Actually, only the observed label set $Y^o$ can be used to establish the prediction model. The naive approach uses $Y^o$ to minimize the following prediction inaccuracy:

$$L_{\text{naive}}(\hat{Y}, Y^o) = \frac{1}{|\mathcal{O}|} \sum_{(u,i) \in \mathcal{O}} e_{u,i} = \frac{1}{|\mathcal{O}|} \sum_{(u,i) \in \mathcal{D}} o_{u,i} e_{u,i}.$$

As mentioned in [11], if the probability of every entry $y_{u,i}$ in $Y$ being missing is identical, then the naive estimator is unbiased, that is $\mathbb{E}_O[L_{\text{naive}}] = L_{\text{real}}$, where $O$ is taken to represent the random variable of observation. The unbiased estimation property of the naive approach is no longer valid when the data is MNAR, which even results in a large difference between $L_{\text{real}}$ and $\mathbb{E}_O[L_{\text{naive}}]$.

**Quantitative Bias-Variance Joint Optimization.** Considering the large difference between $L_{\text{real}}$ and $\mathbb{E}_O[L_{\text{naive}}]$, various unbiased estimation methods have been developed to overcome this problem, such as EIB [10], IPS estimator [6], DR method [11], and various variations of them [5, 7, 8, 12, 13, 15]. The corresponding estimators are given as follows:

$$L_{\text{EIB}}(\hat{Y}, Y^o) = \frac{1}{|\mathcal{D}|} \sum_{(u,i)\in\mathcal{D}} [o_{u,i}e_{u,i} + (1 - o_{u,i})\hat{e}_{u,i}],$$

$$L_{\text{IPS}}(\hat{Y}, Y^o) = \frac{1}{|\mathcal{D}|} \sum_{(u,i)\in\mathcal{D}} \frac{o_{u,i}}{\hat{p}_{u,i}} e_{u,i},$$

$$L_{\text{DR}}(\hat{Y}, Y^o) = \frac{1}{|\mathcal{D}|} \sum_{(u,i)\in\mathcal{D}} \left[ \hat{e}_{u,i} + \frac{o_{u,i}}{\hat{p}_{u,i}}(e_{u,i} - \hat{e}_{u,i}) \right],$$

where $\hat{e}_{u,i} = w|\hat{y}_{u,i} - \gamma|$ for MAE or $\hat{e}_{u,i} = w(\hat{y}_{u,i} - \gamma)^2$ for MSE of missing entries $y_{u,i}$ is the imputed errors, and $\hat{p}_{u,i} \in (0,1)$ is the estimation of the observation propensity, i.e., $p_{u,i} = \Pr(o_{u,i} = 1) \in (0,1)$. Note that $w$ and $\gamma$ are hyper-parameters [10]. For the naive, EIB, IPS, and DR estimators, their biases, variances and generalization bounds are summarized in Table 4 (see Appendix A for more details), where $\Delta_{u,i} = 1 - \frac{p_{u,i}}{\hat{p}_{u,i}}$ and $\delta_{u,i} = e_{u,i} - \hat{e}_{u,i}$. In general, the learning of the imputation model also involves the MNAR problem. Some joint learning algorithms [11, 12] employ the propensity model to overcome this problem. Therefore, propensity estimation has a crucial role in unbiasedness and robustness. Besides, it is difficult to accurately estimate imputed errors for all user-item pair $(u,i)$ in the sense that it is difficult to achieve the unbiasedness of the EIB estimator. If the propensity estimation $\hat{p}_{u,i}$ is accurate, that is $\hat{p}_{u,i} = p_{u,i}$, then IPS and DR estimators are unbiased. For a new dataset, we cannot know in advance the range of the propensities in this dataset. Therefore, a new dataset may introduce extremely small propensities to lead to unbounded variances of IPS and DR, which will disrupt the stability of estimators, especially for larger datasets. It is unacceptable for real industrial scenarios. Specifically, the smaller the propensity, the larger the variance. When the propensity tends to zero, the variance tends to infinity (see Appendix B for more details). Similarly, variances of other IPS-based and DR-based unbiased estimation methods [15] are also unbounded. On the other hand, although the variances of naive and EIB methods are bounded when the prediction error $e_{u,i}$ is bounded, it is difficult and even impossible to achieve an unbiased estimation. Other variance reduction estimation methods [5, 7, 8, 12] are generally biased. According to the expressions of estimators and Table 4, the bias and variance of an estimation are determined by the random variable $O$. We found that slightly relaxing the requirements for unbiasedness will lead to a bounded variance for all propensities. Therefore, the core problem of estimation on MNAR data is the bias-variance joint optimization.

## 3 Fine-Grained Dynamic Framework for Quantitative Bias-Variance Joint Optimization

In this section, we first discuss limitations of regularization techniques and the relationship between unbiasedness of the generalized estimator and its generalization bound, which illustrate the core of the fine-grained estimator design. Then, the dynamic estimation framework for quantitative bias-variance optimization is present. Its generalization bounds and variances are reduced and bounded with theoretical guarantees.

### 3.1 Limitations of Regularization Techniques

Define the general form of the estimator with regularization as

$$L_{\text{Est+Reg}} = \underbrace{\frac{1}{|\mathcal{D}|} \sum_{(u,i)\in\mathcal{D}} \left[ f(o_{u,i}, \hat{p}_{u,i})e_{u,i} + g(o_{u,i}, \hat{p}_{u,i})\hat{e}_{u,i} \right]}_{L_{\text{Est}}} + \lambda \underbrace{\frac{1}{|\mathcal{D}|} \sum_{(u,i)\in\mathcal{D}} h(o_{u,i}, \hat{p}_{u,i})}_{L_{\text{Reg}}}, \tag{1}$$

where $f(\cdot, \cdot) \neq 0$ with $f(0, \hat{p}_{u,i}) = 0$, $g(\cdot, \cdot)$, and $h(\cdot, \cdot)$ are functions with respect to $o$ and $\hat{p}$. $L_{\text{Est}}$ and $L_{\text{Reg}}$ are prediction inaccuracies of the estimator and regularizer, respectively. For all $(u, i)$ pairs, they satisfy $f(o_{u,i}, \hat{p}_{u,i})e_{u,i} + g(o_{u,i}, \hat{p}_{u,i})\hat{e}_{u,i} \geq 0$ and $h(o_{u,i}, \hat{p}_{u,i}) \geq 0$. $\lambda > 0$ is a scalar weight. The generalized estimator form $L_{\text{Est}}$ given in Eq. (1) covers the vast majority of existing estimators involving EIB [10], IPS [6], DR [11], More Robust DR (MRDR) [5], Targeted DR (TDR) [15], MIS [16], IPS/DR-SV [16], and other IPS-based and DR-based methods. On the other hand, almost all existing regularization designs, including the Sample Variance (SV) [16], mean inverse square (MIS) [16], Balancing-Mean-Square Error (BMSE) [8], and so forth, can be transformed into the form $L_{\text{Reg}}$ given in (1). In previous works, the regularization technique plays a critical role in variance reduction of estimators and improvement of the generalization performance to a certain extent. However, it still have some inevitable limitations described in the following box.

---

*Core Results*

1) *For the general estimator with regularization $L_{Est+Reg}$, it is impossible to reduce variance without sacrificing unbiasedness.*

2) *Regularization $L_{Reg}$ cannot guarantee a bounded variance and generalization bound.*

---

In what follows, we provide a detailed theoretical analysis to reveal the aforementioned limitations of the regularization technique. Considering the variance of $L_{\text{Est+Reg}}$, we have

$$\mathbb{V}_O[L_{\text{Est+Reg}}] = \mathbb{V}_O[L_{\text{Est}}] + 2\lambda \text{Cov}(L_{\text{Est}}, L_{\text{Reg}}) + \lambda^2 \mathbb{V}_O[L_{\text{Reg}}].$$

As mentioned in [8], when the parameter $\lambda$ is set as the optimal parameter $\lambda_{\text{opt}} = -\frac{\text{Cov}(L_{\text{Est}}, L_{\text{Reg}})}{\mathbb{V}_O[L_{\text{Reg}}]}$, the variance $\mathbb{V}_O[L_{\text{Est+Reg}}]$ achieves its minimum and satisfies $\mathbb{V}_O[L_{\text{Est+Reg}}] \leq \mathbb{V}_O[L_{\text{Est}}]$ in the sense that the regularization term $\lambda L_{\text{Reg}}$ enables the estimator $L_{\text{Est+Reg}}$ to reduce its variance. However, the covariance $\text{Cov}(L_{\text{Est}}, L_{\text{Reg}})$ needs to fulfill $\text{Cov}(L_{\text{Est}}, L_{\text{Reg}}) < 0$ as $\lambda > 0$. Otherwise, an inappropriate parameter will result in an increased variance. The formal theoretical results are provided by Theorems 3.1 and Corollary 3.2, which reveal the limitation 1) (see Appendix D for proofs). Corollary 3.2 is the contrapositive of Theorems 3.1.

**Theorem 3.1.** *Let $L_{Est+Reg}$ be defined in (1) and the estimator $L_{Est}$ be unbiased. If $L_{Est+Reg}$ is unbiased, then the variance of $L_{Est+Reg}$ is greater than the variance of the original estimator $L_{Est}$.*

**Corollary 3.2.** *If the variance of $L_{Est+Reg}$ is less than the variance of the original estimator $L_{Est}$, then $L_{Est+Reg}$ is not unbiased.*

We further find that, if the variance of the original estimator is unbounded, the variances of estimators cannot be bounded by introducing a regularizer even if $\hat{p}_{u,i} = p_{u,i}$. The theoretical results are shown in Theorem 3.3(see Appendix D for proofs).

**Theorem 3.3.** *Let the bias of $L_{Est+Reg}$ be bounded and the variance of $L_{Est}$ satisfy $\lim_{p_{u,i} \to 0} \mathbb{V}_O[L_{Est}|\hat{p}_{u,i} = p_{u,i}] = \infty$. Then, there exists no regularizer $L_{Reg}$ that enables the variance and generalization bound of the estimator bounded even the learned imputed errors or propensities are accurate.*

According to the previous works and the present Theorem 3.3, regularizers enable variance reduction to a certain extent while cannot enable estimators to possess bounded variances and generalization bounds. In other words, regularization techniques have limited impact on improving the predictive performance of the model. In the next subsection, a novel perspective of dynamic estimator designs is proposed, which not only achieves quantitative bias-variance joint optimization but also guarantees bounded variances and generalization bounds.

### 3.2 Dynamic Estimator Designs With Quantitative Optimization

Most of the existing estimators are based on IPS and DR methods, which are elaborately designed to reduce bias or variance. However, all these estimators are static estimators in the sense that they cannot achieve bias-variance joint optimization for each user-item pair $(u, i)$. Even though some methods [5, 8] can effectively reduce the variance of estimators, the estimators are biased and the corresponding variances are unbounded. In this subsection, the core results are provided in the following box. Also, based on theses results, we develop a fine-grained dynamic framework with

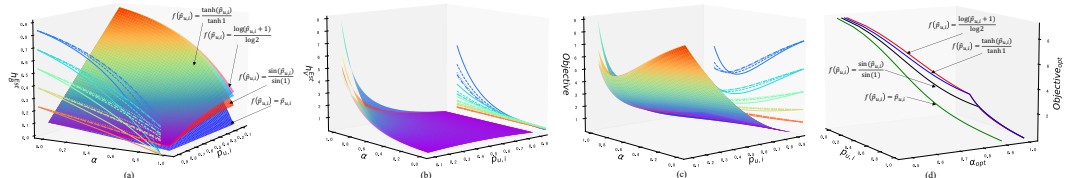

Figure 1: The surfaces of determining factors and the objective function of dynamic estimators, and the optimal objective values: (a) $h_B^{\text{Est}}$; (b) $h_V^{\text{Est}}$; (c) $w_1(h_B^{\text{Est}})^2 + w_2 h_V^{\text{Est}}$; (d) Objective$_{\text{opt}}$.

quantitative optimization to guarantee the reduction and boundedness of variances and generalization bounds

---

*Core Results*

3) For the generalized estimator $L_{\text{Est}}$, unbiasedness of the estimator will inevitably lead to the unbounded variance and generalization bound.

4) The core of the estimator design involves not merely a simple bias-variance trade-off, but rather a quantitative joint optimization of both bias and variance.

---

We find that the unbiased estimators with general form generally possess unbounded variances, which is formally derived in Theorem 3.4. Its proofs are provided in Appendix D.

**Theorem 3.4** (Limitation of Static Estimator). *Given prediction errors $e_{u,i}$, imputed errors $\hat{e}_{u,i}$, and learned propensities $\hat{p}_{u,i}$ for all user-item pairs $(u,i)$, if for any $e_{u,i} - g(0, \hat{p}_{u,i})\hat{e}_{u,i} \neq 0$, $L_{Est}$ given in (1) is unbiased, then the corresponding variance and generalization bound are unbounded.*

According to Theorem 3.4, the core objective of estimators is not merely about eliminating bias, reducing variance, or simply achieving a bias-variance trade-off but about a quantitative joint optimization between bias and variance. Therefore, as mentioned in *Core Results*, it is necessary to develop a dynamic estimation framework to achieve the quantitative joint optimization.

**Design Principle of Dynamic Estimators.** The IPS-based and DR-based dynamic learning frameworks are designed as

$$L_{\text{D-IPS}} = \frac{1}{|\mathcal{D}|} \sum_{(u,i) \in \mathcal{D}} \frac{o_{u,i}}{f^{\alpha_{u,i}}(\hat{p}_{u,i})} e_{u,i}, \; L_{\text{D-DR}} = \frac{1}{|\mathcal{D}|} \sum_{(u,i) \in \mathcal{D}} \left( \hat{e}_{u,i} + \frac{o_{u,i}}{f^{\alpha_{u,i}}(\hat{p}_{u,i})} \delta_{u,i} \right), \quad (2)$$

where $f(\cdot)$ is a designed function and $\alpha_{u,i} \in [0,1]$ is optimizable parameters. When $f(\hat{p}_{u,i}) = \hat{p}_{u,i}$ and $\forall \alpha_{u,i} = 1$, D-IPS and D-DR are equivalent to the original IPS and DR estimators, respectively, which possess unbiasedness. When $f(\hat{p}_{u,i}) = \hat{p}_{u,i}$ and $\forall \alpha_{u,i} = 0$, D-IPS and D-DR are equivalent to $\frac{|\mathcal{O}|}{|\mathcal{D}|} L_{\text{naive}}$ and EIB estimators, which have bounded variances and generalization bounds. The function $f(\hat{p}_{u,i})$ in (2) is actually a mapping, which balances the bias and variance of estimators. The design principles of $f(\hat{p}_{u,i})$ are provided as follows:

- **(Isotonic Propensity)** $f(\hat{p}_{u,i})$ with $f(0) = 0$, $f(1) = 1$, and $f(\hat{p}_{u,i}) > \hat{p}_{u,i}$ is a monotonically increasing function.

- **(Same Order)** $\lim_{\hat{p}_{u,i} \to 0} \frac{\hat{p}_{u,i}}{f(\hat{p}_{u,i})} = C$, where $C$ is a positive constant.

Some specific expressions of $f(\hat{p}_{u,i})$ fulfilling the above design principles are summarized in Table 1. The corresponding biases, variances and tail bounds of D-IPS and D-DR estimators are formally formulated in Lemmas D.1–D.4 given in Appendix D. From the biases and variances of the D-IPS and D-DR methods given as

$$\text{Bias}(L_{\text{D-IPS}}) = \frac{1}{|\mathcal{D}|} \left| \sum_{(u,i) \in \mathcal{D}} h_B^{\text{Est}}(\hat{p}_{u,i}, p_{u,i}, \alpha_{u,i}) e_{u,i} \right|, \; \text{Bias}(L_{\text{D-DR}}) = \frac{1}{|\mathcal{D}|} \left| \sum_{(u,i) \in \mathcal{D}} h_B^{\text{Est}}(\hat{p}_{u,i}, p_{u,i}, \alpha_{u,i}) \delta_{u,i} \right|,$$

$$\mathbb{V}_O[L_{\text{D-IPS}}] = \frac{1}{|\mathcal{D}|^2} \sum_{(u,i) \in \mathcal{D}} h_V^{\text{Est}}(\hat{p}_{u,i}, p_{u,i}, \alpha_{u,i}) e_{u,i}^2, \; \mathbb{V}_O[L_{\text{D-DR}}] = \frac{1}{|\mathcal{D}|^2} \sum_{(u,i) \in \mathcal{D}} h_V^{\text{Est}}(\hat{p}_{u,i}, p_{u,i}, \alpha_{u,i}) \delta_{u,i}^2,$$

Table 1: The specific expressions of $f^{\alpha_{u,i}}(\hat{p}_{u,i})$ and their determining factors of the bias and variance.

| $f(\hat{p}_{u,i})$ | $\hat{p}_{u,i}$ | $\frac{\sin(\hat{p}_{u,i})}{\sin(1)}$ | $\frac{\log(\hat{p}_{u,i}+1)}{\log(2)}$ | $\frac{\tanh(\hat{p}_{u,i})}{\tanh(1)}$ |
|---|---|---|---|---|
| $h_B$ | $1-\frac{p_{u,i}}{\hat{p}_{u,i}^{\alpha_{u,i}}}$ | $1-\frac{p_{u,i}\sin^{\alpha_{u,i}}(1)}{\sin^{\alpha_{u,i}}(\hat{p}_{u,i})}$ | $1-\frac{p_{u,i}\log^{\alpha_{u,i}}(2)}{\log^{\alpha_{u,i}}(\hat{p}_{u,i}+1)}$ | $1-\frac{p_{u,i}\tanh^{\alpha_{u,i}}(1)}{\tanh^{\alpha_{u,i}}(\hat{p}_{u,i})}$ |
| $h_V$ | $\frac{p_{u,i}(1-p_{u,i})}{\hat{p}_{u,i}^{2\alpha_{u,i}}}$ | $\frac{p_{u,i}(1-p_{u,i})\sin^{2\alpha_{u,i}}(1)}{\sin^{2\alpha_{u,i}}(\hat{p}_{u,i})}$ | $\frac{p_{u,i}(1-p_{u,i})\log^{2\alpha_{u,i}}(2)}{\log^{2\alpha_{u,i}}(\hat{p}_{u,i}+1)}$ | $\frac{p_{u,i}(1-p_{u,i})\tanh^{2\alpha_{u,i}}(1)}{\tanh^{2\alpha_{u,i}}(\hat{p}_{u,i})}$ |

where $h_B^{\text{Est}}(\hat{p}_{u,i},p_{u,i},\alpha_{u,i})=1-\frac{p_{u,i}}{f^{\alpha_{u,i}}(\hat{p}_{u,i})}$ and $h_V^{\text{Est}}(\hat{p}_{u,i},p_{u,i},\alpha_{u,i})=\frac{p_{u,i}(1-p_{u,i})}{f^{2\alpha_{u,i}}(\hat{p}_{u,i})}$, functions $h_B^{\text{Est}}$ and $h_V^{\text{Est}}$ determine the biases and variance, respectively. $h_B^{\text{Est}}$ and $h_V^{\text{Est}}$ corresponding the specific expressions of $f(\hat{p}_{u,i})$ are given in Table 1. The monotonicity of bias and variance are provided in Appendix D Proposition D.3. The surfaces of $h_B^{\text{Est}}$ and $h_V^{\text{Est}}$ are plotted in Figs. 1(a) and (b). It is observed that $h_B^{\text{Est}}$ is monotonically decreasing and $h_V^{\text{Est}}$ is monotonically increasing as the number of $\alpha_{u,i}$ increases.

**Bias-Variance Quantitative Joint Optimization.** According to Proposition D.3 given in Appendix D, the bias-variance trade-off problem can be quantitatively formalized as the following joint optimization problem:

$$\text{Objective} = \min_{\alpha_{u,i}}\left\{w_1\text{Bias}(L(\alpha_{u,i}))+w_2\mathbb{V}_O[L(\alpha_{u,i})]\right\}, \text{ s.t. } 0\le\alpha_{u,i}\le1, \tag{3}$$

where $w_1$ and $w_2$ are weights of the bias and variance.

According to determine factors $h_B^{\text{Est}}$ and $h_V^{\text{Est}}$ of bias and variance, respectively, the bias-variance joint optimization problem can be defined as

$$\text{Objective}^{\text{opt}} = \min_{\alpha_{u,i}}\left\{w_1 E_B(h_B^{\text{Est}}(\alpha_{u,i}))+w_2 E_V(h_V^{\text{Est}}(\alpha_{u,i}))\right\}, \text{ s.t. } 0\le\alpha_{u,i}\le1. \tag{4}$$

For each user-item pair $(u,i)$, minimizing $E_B(h_B^{\text{Est}})$ and $E_V(h_V^{\text{Est}})$ given accurate propensity estimations $\hat{p}_{u,i}$ leads to the bias and variance reduction, respectively. Therefore, the optimal parameter $\alpha_{u,i}\in[0,1]$ for each user-item pair $(u,i)$ can achieve fine-grained bias-variance joint optimization. The function $h_B^{\text{Est}}$ under $\alpha_{u,i}\in[0,1]$, $f(\hat{p}_{u,i})\ge\hat{p}_{u,i}$ and $\hat{p}_{u,i}=p_{u,i}$ satisfies $h_B^{\text{Est}}(\hat{p}_{u,i},p_{u,i},\alpha_{u,i})\in[0,1)$. On the other hand, $h_V^{\text{Est}}$ under $\alpha_{u,i}\in[0,1]$ and $\hat{p}_{u,i}=p_{u,i}$ satisfies $h_V^{\text{Est}}(\hat{p}_{u,i},p_{u,i},\alpha_{u,i})\in[0,\infty)$. Therefore, the objective function in (4) can be simplified as $w_1 h_B^{\text{Est}}(\alpha_{u,i})+w_2 h_V^{\text{Est}}(\alpha_{u,i})$. The curves of objective functions under different designed functions $f(\cdot)$ are given in Fig. 1(c). It can be observed that for a fixed propensity, there exists an $\alpha$ such that the objective function attains the minimum value. Besides, different measure metrics are also applicable for dynamic estimators, such as $E(h^{\text{Est}})=(h^{\text{Est}}(\alpha_{u,i}))^2$, $E(h^{\text{Est}}(\alpha_{u,i}))=\ln(\cosh(h^{\text{Est}}(\alpha_{u,i})))$, and so on. In what follows, under the objective function $w_1 h_B^{\text{Est}}+w_2 h_V^{\text{Est}}$, the analytical solution of the optimal parameter $\alpha_{u,i}^{\text{opt}}$ is given in Theorem 3.5 (see Appendix D for proofs).

**Theorem 3.5** (The optimal parameter $\alpha_{u,i}^{\text{opt}}$). *Let the learned propensities be accurate, i.e., $\hat{p}_{u,i}=p_{u,i}$. For weights $w_1$ and $w_2$, the objective function $w_1 h_B^{Est}+w_2 h_V^{Est}$ under $\alpha_{u,i}\in[0,1]$ achieves its minimum at*

$$\alpha_{u,i}^{opt} = \min\left\{\max\left\{\frac{\ln\left(\frac{2w_2}{w_1}(1-p_{u,i})\right)}{\ln(f(p_{u,i}))},0\right\},1\right\}. \tag{5}$$

From the expression of the optimal parameter (5), the optimal solution of (4) under different weights depends on the weight ratio $w_2/w_1$. Under different designed function $f(\cdot)$, the schematic diagram of optimal objective values corresponding to the optimal parameter $\alpha_{u,i}^{opt}$ is shown in Fig. 1(d). Next, the generalization bounds of the developed dynamic estimator framework are further discussed. The formalized results are derived in Theorem 3.6 (see Appendix D for more details).

**Theorem 3.6** (Generalization Bounds of D-IPS and D-DR). *For any finite hypothesis space $\mathcal{H}$ of $\hat{Y}$ and the optimal prediction matrix $\hat{Y}^-$, given $\hat{e}_{u,i}$ and $\hat{p}_{u,i}$ for all $(u,i)\in\mathcal{D}$, with probability $1-\rho$, the prediction inaccuracies $L_{D\text{-}IPS}(\hat{Y}^-,Y)$ and $L_{D\text{-}DR}(\hat{Y}^-,Y)$ under D-IPS and D-DR have*

*the following upper bounds*

$$L_{D\text{-}IPS}(\hat{Y}^-, Y^O) + \underbrace{\sum_{(u,i)\in\mathcal{D}} \frac{|h_B^{Est}(\hat{p}_{u,i}, p_{u,i}, \alpha_{u,i})e_{u,i}^-|}{|\mathcal{D}|}}_{\text{Bias Term}} + \underbrace{h_G^{Est}(e_{u,i}^+)}_{\text{Variance Term}},$$

$$L_{D\text{-}DR}(\hat{Y}^-, Y^O) + \underbrace{\sum_{(u,i)\in\mathcal{D}} \frac{|h_B^{Est}(\hat{p}_{u,i}, p_{u,i}, \alpha_{u,i})\delta_{u,i}^-|}{|\mathcal{D}|}}_{\text{Bias Term}} + \underbrace{h_G^{Est}(\delta_{u,i}^+)}_{\text{Variance Term}},$$

*where $e_{u,i}^+$ and $\delta_{u,i}^+$ are the error and error deviation corresponding to $\hat{Y}^+ = \arg\max_{\hat{Y}\in\mathcal{H}}\left\{\sum_{(u,i)\in\mathcal{D}}\left(\frac{e_{u,i}}{f^{\alpha_{u,i}}(\hat{p}_{u,i})}\right)^2\right\}$ and $\hat{Y}^+ = \arg\max_{\hat{Y}\in\mathcal{H}}\left\{\sum_{(u,i)\in\mathcal{D}}\left(\frac{\delta_{u,i}}{f^{\alpha_{u,i}}(\hat{p}_{u,i})}\right)^2\right\}$, respectively, and the function $h_G^{Est}$ is formulated as $h_G^{Est}(z_{u,i}^+) = \sqrt{\frac{\log(\frac{2|\mathcal{H}|}{\rho})}{2|\mathcal{D}|^2}\sum_{(u,i)\in\mathcal{D}}\left(\frac{z_{u,i}^+}{f^{\alpha_{u,i}}(\hat{p}_{u,i})}\right)^2}$*

From Theorem 3.6, the bias-variance joint optimization is actually to minimize generalization bounds, which include both the bias term and the variance term. Besides, the dynamic estimators with the optimal parameter $\alpha_{u,i}^{opt}$ make variances and generalization bounds bounded. The formal result is given in Theorem 3.7 (The corresponding proofs and bounds of variances are given in Appendix D).

**Theorem 3.7** (Boundedness of Variance and Generalization Bound). *Let $\alpha_{u,i}^{opt} \in [0,1]$ be the optimal parameter of (4). If the dynamic estimators adopt $\alpha_{u,i}^{opt}$ as the parameter, then the corresponding variance and generalization bounds are bounded.*

## 4  Experiments

In this section, we conduct extensive experiments to compare the performance of the present dynamic learning framework with existing SOTA approaches and to answer the following questions: **Q1**: Does the developed dynamic learning framework improve the prediction performance compared with the SOTA approaches? **Q2**: Do the present dynamic estimator designs reduce the variance and make performance more stable compared with the SOTA approaches? **Q3**: How do the performance and variance of the proposed method change under different optimization weights and estimator functions?

### 4.1  Experimental Setup

**Dataset and Preprocessing.**  Three real-world datasets with MNAR and MAR samples are used to conduct the experiments, namely COAT with 4,640 MAR and 6,960 MNAR ratings of 290 users to 300 coats, YAHOO! R3 with 54,000 MAR and 311,704 MNAR ratings of 15,400 users to 1,000 songs, and KUAIREC with 4,676,570 video watching ratio records of 1,411 users to 3,327 video. Similar to literature [8, 7, 5], the rating scores in COAT and YAHOO! R3 are binarized as 1 when it is greater than three, otherwise as 0. For the KUAIREC dataset, the video watching ratios are binarized as 0 when it is less than two, otherwise as 1.

**Baselines and Experimental Details.**  To avoid the uncertainty caused by the prediction and observe the performance of the prediction model, we take the matrix factorization (MF) [17] as the base model and compare the present dynamic learning framework with the following representative IPS-based and DR-based approaches: naive **MF** [17], **IPS** [6], **SNIPS** [13], **IPS-AT** [18], **CVIB** [19], **IPS-V2** [8], **DR** [11], **DR-JL** [11], **MRDR-JL** [5], **Stable DR** [12], **Stable MRDR** [12], **TDR-CL** [15], **TMRDR-CL** [15], **DR-V2** [8]. Here, we adopt the two common metrics used in recommender system, i.e., area under the ROC curve (AUC), and normalized discounted cumulative gain (NDCG), to evaluate the performance of prediction models. To guarantee the fair comparison, we set the same parameters for all approaches. The learning rates are tuned in $\{0.001, 0.005, 0.01, 0.05\}$ and weight decay is tuned in $\{1, 1e-1, 1e-2, 1e-3, 1e-4, 1e-5, 1e-6\}$. Note that, for *XX* and *D-XX* approaches, their model structures and parameters are identical. Every approach is preformed 10 times to record its mean and standard deviation.

## 4.2 Performance Comparison (for Q1 and Q2)

The MNAR records in datasets are used to train the prediction model and the MAR data is employed to evaluate the present dynamic learning approaches and the existing SOTA approaches. The function in proposed *D-XX* approaches is select as $\left(\frac{\log(\hat{p}_{u,i}+1)}{\log(2)}\right)^{\alpha}$. The performance of other functions provided in Table 1 are discussed in subsection 4.3. The performances of the developed dynamic estimators and the SOTA approaches are shown in Table 2, where $\text{Gain}_{\text{AUC}} = (\text{AUC}_{\text{XX}} - \text{AUC}_{\text{D-XX}})/\text{AUC}_{\text{XX}}$ and $\text{Gain}_{\text{N}} = (\text{NDCG}_{\text{XX}} - \text{NDCG}_{\text{D-XX}})/\text{NDCG}_{\text{XX}}$, e.g. $\text{Gain}_{\text{AUC}} = (\text{AUC}_{\text{IPS}} - \text{AUC}_{\text{D-IPS}})/\text{AUC}_{\text{IPS}}$, $\text{Gain}_{\text{N}} = (\text{NDCG}_{\text{IPS}} - \text{NDCG}_{\text{D-IPS}})/\text{NDCG}_{\text{IPS}}$. The weights in bias-variance joint optimization are set as $w_1 = 1$ and $w_2 = 0.1$. For almost all of metrics and datasets, the performances of IPS, IPS-AT, CVIB, IPS-V2, DR, DR-JL, MRDR, DR-V2 outperform the naive method while the naive approach has smaller variance, which implies that unbiased estimators possess high variance. Besides, it can be observed that estimators with the dynamic learning mechanism greatly improve the performances and reduce the variances of various debiased approaches, such as IPS and D-IPS, DR and D-DR, DR-JL and D-DR-JL. Meanwhile, for SNIPS, MRDR-JL, the variance reduction of dynamic estimators do not seem obvious. One possible reason is that these approaches themselves can effectively reduce the variances of estimators by sacrificing unbiasedness. These experiment results further verify Theorem 3.4.

Table 2: Performances of the proposed method and baselines (mean $\pm$ standard deviation across 10 runs).

| | Coat | | | | Yahoo! R3 | | | | KuaiRec | | | |
|---|---|---|---|---|---|---|---|---|---|---|---|---|
| Methods | AUC | Gain$_{\text{AUC}}$ | NDCG@5 | Gain$_{\text{N}}$ | AUC | Gain$_{\text{AUC}}$ | NDCG@5 | Gain$_{\text{N}}$ | AUC | Gain$_{\text{AUC}}$ | NDCG@50 | Gain$_{\text{N}}$ |
| naive | 0.7429±0.0046 | – | 0.6173±0.0065 | – | 0.6619±0.0011 | – | 0.6798±0.0019 | – | 0.7498±0.0010 | – | 0.7356±0.0012 | – |
| IPS | 0.7539±0.0058 | – | 0.6496±0.0093 | – | 0.6624±0.0025 | – | 0.6583±0.0020 | – | 0.7314±0.0023 | – | 0.7450±0.0015 | – |
| SNIPS | 0.7423±0.0038 | – | 0.6110±0.0056 | – | 0.6727±0.0022 | – | 0.6647±0.0019 | – | 0.8015±0.0020 | – | 0.8082±**0.0007** | – |
| IPS-AT | 0.7692±0.0023 | – | 0.6290±0.0069 | – | 0.6570±0.0072 | – | 0.6720±0.0019 | – | 0.7733±0.0063 | – | 0.8003±0.0037 | – |
| CVIB | 0.7448±0.0032 | – | 0.6123±0.0082 | – | 0.6119±0.0020 | – | 0.6766±0.0017 | – | 0.7727±0.0064 | – | 0.7852±0.0065 | – |
| IPS-V2 | 0.7737±0.0024 | – | 0.6514±0.0056 | – | 0.6656±0.0022 | – | 0.6434±0.0026 | – | 0.7787±0.0016 | – | 0.7905±0.0029 | – |
| **D-IPS (Ours)** | 0.7777±0.0015 | 3.16% | 0.6584±0.0049 | 1.35% | 0.6767±0.0024 | 2.16% | 0.6630±0.0027 | 0.71% | 0.7947±**0.0005** | 8.65% | 0.7876±0.0009 | 5.71% |
| **D-SNIPS (Ours)** | 0.7429±0.0036 | 0.08% | 0.6096±0.0062 | -0.23% | **0.7018**±0.0012 | 4.33% | 0.6899±0.0025 | 3.79% | **0.8026**±0.0017 | 0.137% | 0.8084±0.0005 | 0.247% |
| **D-IPS-AT (Ours)** | 0.7705±0.0012 | 0.17% | 0.6367±0.0052 | 1.22% | 0.6913±0.0029 | 4.33% | 0.6769±0.0047 | 0.73% | 0.7882±0.0042 | 1.93% | **0.8143**±0.0023 | 1.75% |
| DR | 0.7538±0.0032 | – | 0.6425±0.0096 | – | 0.6863±0.0013 | – | 0.6738±0.0033 | – | 0.7701±0.0058 | – | 0.7818±0.0029 | – |
| DR-JL | 0.7574±0.0046 | – | 0.6496±0.0141 | – | 0.6853±0.0012 | – | 0.6707±0.0019 | – | 0.7808±0.0034 | – | 0.7930±0.0033 | – |
| MRDR-JL | 0.7590±0.0031 | – | 0.6502±0.0074 | – | 0.6851±0.0017 | – | 0.6708±0.0024 | – | 0.7735±0.0008 | – | 0.8121±0.0013 | – |
| Stable DR | 0.7648±0.0013 | – | 0.6315±0.0040 | – | 0.6925±0.0019 | – | 0.6749±0.0023 | – | 0.7812±0.0007 | – | 0.7928±0.0040 | – |
| Stable MRDR | 0.7645±**0.0009** | – | 0.6318±**0.0025** | – | 0.6929±0.0016 | – | 0.6753±0.0011 | – | 0.7844±0.0013 | – | 0.7752±0.0021 | – |
| TDR-CL | 0.7639±0.0032 | – | 0.6541±0.0102 | – | 0.6797±**0.0006** | – | 0.6842±0.0011 | – | 0.7858±0.0016 | – | 0.7776±0.0015 | – |
| TMRDR-CL | 0.7690±0.0016 | – | 0.6363±0.0041 | – | 0.6877±0.0016 | – | 0.6877±**0.0009** | – | 0.7801±0.0017 | – | 0.8047±0.0013 | – |
| DR-V2 | 0.7749±0.0024 | – | 0.6625±0.0092 | – | 0.6846±0.0043 | – | 0.6613±0.0054 | – | 0.7839±0.0027 | – | 0.7923±0.0056 | – |
| **D-DR (Ours)** | **0.7804**±0.0023 | 3.53% | **0.6671**±0.0051 | 3.83% | **0.7043**±0.0026 | 1.98% | **0.7043**±0.0042 | 4.53% | 0.7835±0.0040 | 3.31% | 0.7835±0.0040 | 2.17% |
| **D-DR-JL (Ours)** | 0.7775±0.0016 | 2.65% | 0.6577±0.0036 | 1.25% | 0.6913±0.0014 | 0.88% | 0.6721±0.0028 | 0.21% | 0.7742±0.0011 | -0.845% | 0.7897±0.0017 | -0.416% |
| **D-MRDR-JL (Ours)** | 0.7786±0.0025 | 2.58% | 0.6616±0.0044 | 1.75% | 0.6917±0.0027 | 0.96% | 0.6735±0.0038 | 0.40% | 0.7918±0.0011 | 2.37% | 0.8105±0.0010 | -0.197% |

## 4.3 Ablation Studies (for Q3)

**Effects of Different Weights and Functions in Dynamic Estimators.** In subsection 3.2, we provide four specific dynamic estimators given in Table 1. We set the weights in bias-variance joint optimization (4) as $w_1 = 1$ and $w_2 = [0.02, 0.04, 0.06, 0.08, 1]$ for these four dynamic estimators to investigate the effects of weights on the performances and variances. From Eq. (5), the optimal parameter $\alpha_{\text{opt}}$ is determined by the ratio $\frac{w_2}{w_1}$ and $\alpha_{\text{opt}}$ determines the objective function. Therefore, we just focus on the effects of the weight ratios on the performance and variances of estimators, which are given in Fig. 2. It can be observed that, for D-IPS, D-IPS-AT, D-DR, D-DR-JL, and D-MRDR-JL approaches, the performances increased at first and then decreased as the number of the weight ratio increases. Meanwhile, the variances seem to achieve their minimums when the performances achieve their highest values. Since the smaller the weight ratio is, the smaller the bias of the dynamic estimator is, the experimental results given in Fig. 2 reveal that the unbiasedness of estimators is not exactly equivalent to the performances of estimators. Actually, from the generalization bounds given in Theorem 3.6, the bias-variance joint optimization enable estimators to minimize the generalization

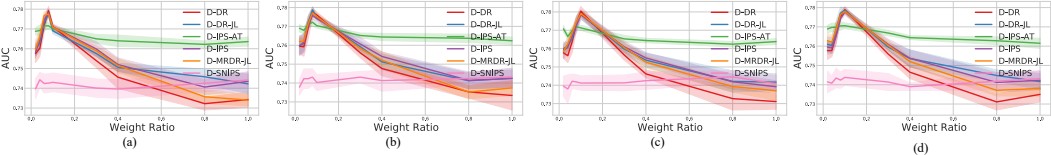

Figure 2: Effects of different weight ratios $\frac{w_2}{w_1}$ on performances of dynamic estimators under different functions $f^{\alpha}(\hat{p}_{u,i})$: (a) $\hat{p}_{u,i}^{\alpha}$; (b) $\left(\frac{\sin(\hat{p}_{u,i})}{\sin(1)}\right)^{\alpha}$; (c) $\left(\frac{\log(\hat{p}_{u,i}+1)}{\log(2)}\right)^{\alpha}$; (d) $\left(\frac{\tanh(\hat{p}_{u,i})}{\tanh(1)}\right)^{\alpha}$.

Table 3: Effects of different functions on performances of dynamic estimators.

| $f^{\alpha}(\hat{p}_{u,i})$ | $\hat{p}_{u,i}^{\alpha}$ | | | | $\left(\frac{\sin(\hat{p}_{u,i})}{\sin(1)}\right)^{\alpha}$ | | | |
|---|---|---|---|---|---|---|---|---|
| Methods | AUC | $\text{Gain}_{\text{AUC}}$ | NDCG@5 | $\text{Gain}_{\text{N}}$ | AUC | $\text{Gain}_{\text{AUC}}$ | NDCG@5 | $\text{Gain}_{\text{N}}$ |
| D-IPS | 0.7702±**0.0011** | 2.16% | 0.6362±**0.0043** | -2.06% | 0.7753±0.0017 | 2.84% | 0.6475±0.0043 | -0.32% |
| D-SNIPS | 0.7413±0.0045 | -0.13% | **0.6146**±0.0079 | 0.59% | 0.7392±0.0038 | -0.42% | 0.6109±0.0089 | -0.02% |
| D-IPS-AT | 0.7711±0.0016 | 0.25% | 0.6360±0.0051 | 1.11% | 0.7710±0.0022 | 0.23% | 0.6346±0.0036 | 0.89% |
| D-DR | 0.7710±**0.0014** | 2.28% | 0.6384±**0.0047** | -0.64% | 0.7763±0.0021 | 2.98% | 0.6516±0.0052 | 1.42% |
| D-DR-JL | 0.7695±0.0013 | 1.60% | 0.6346±0.0058 | -2.31% | 0.7748±0.0012 | 2.30% | 0.6444±0.0053 | -0.80% |
| D-MRDR-JL | 0.7711±0.0016 | 1.59% | 0.6365±0.0040 | -2.11% | 0.7751±**0.0012** | 2.12% | 0.6470±**0.0038** | -0.49% |

| $f^{\alpha}(\hat{p}_{u,i})$ | $\left(\frac{\log(\hat{p}_{u,i}+1)}{\log(2)}\right)^{\alpha}$ | | | | $\left(\frac{\tanh(\hat{p}_{u,i})}{\tanh(1)}\right)^{\alpha}$ | | | |
|---|---|---|---|---|---|---|---|---|
| Methods | AUC | $\text{Gain}_{\text{AUC}}$ | NDCG@5 | $\text{Gain}_{\text{N}}$ | AUC | $\text{Gain}_{\text{AUC}}$ | NDCG@5 | $\text{Gain}_{\text{N}}$ |
| D-IPS | **0.7777**±0.0015 | 3.16% | **0.6584**±0.0049 | 1.35% | 0.7771±0.0016 | 3.08% | 0.6578±0.0048 | 1.26% |
| D-SNIPS | **0.7429**±**0.0036** | 0.08% | 0.6096±**0.0062** | -0.23% | 0.7418±0.0070 | -0.07% | 0.6115±0.0082 | 0.08% |
| D-IPS-AT | 0.7705±0.0012 | 0.17% | **0.6367**±0.0052 | 1.22% | **0.7718**±**0.0011** | 0.34% | 0.6357±**0.0029** | 1.07% |
| D-DR | **0.7804**±0.0023 | 3.53% | **0.6671**±0.0051 | 3.83% | 0.7792±0.0019 | 3.37% | 0.6608±0.0053 | 2.85% |
| D-DR-JL | 0.7775±0.0016 | 2.65% | **0.6577**±**0.0036** | 1.25% | **0.7782**±**0.0011** | 2.75% | 0.6537±0.0039 | 0.63% |
| D-MRDR-JL | **0.7786**±0.0025 | 2.58% | **0.6616**±0.0044 | 1.75% | 0.7779±0.0017 | 2.49% | 0.6576±0.0071 | 1.14% |

bounds and then further improve the generalization performance. Meanwhile, we find that the variances of dynamic estimators is not decreasing when the weight ratio increases. This because, for different ratios, the global minimum of the objective function (4) cannot be reached within the interval $\alpha \in [0,1]$. For SNIPS, the property of variance reduction might lead to the non-obvious performance and variance trends.

Under the identical weight ratio $\frac{w_2}{w_1} = 0.1$, we further discuss the effects of different functions $f^{\alpha}(\hat{p}_{u,i})$ on the prediction performance and variance. The experimental results are shown in Fig. 3. Nearly all dynamic estimators with different function expressions outperform the corresponding debiased approaches given in Table 3. It further demonstrates that the proposed dynamic learning mechanism can greatly improve the performance of the original estimator. Besides, the prediction performance of the dynamic estimator with $f^{\alpha}(\hat{p}_{u,i}) = \left(\frac{\log(\hat{p}_{u,i}+1)}{\log(2)}\right)^{\alpha}$ outperforms other dynamic estimators.

## 5    Related Work

Aiming at the prediction model bias caused by the MNAR data, EIB [10] and IPS [6] approaches are two classical unbiased estimators. To leverage the advantages of EIB and IPS, the DR method [11] was designed to make the unbiasedness of estimator doubly robust. Focusing on the unbiasedness of estimators, various estimation methods have been proposed to overcome mixed or even unknown biases in the data [9], to solve the sample selection bias problem in the multi-task learning [20, 21, 22, 23], to improve the performance of the propensity model by different approaches [18, 14], and so forth. A multiple robust estimator is developed in [24] by taking the advantage of multiple candidate imputation and propensity models, which is unbiased when any of the imputation or propensity models, or a linear combination of these models is accurate. From a novel function balancing perspective, Li et al. propose to approximate the balancing functions in reproducing kernel Hilbert space [25]. Moreover, aimed at limitations of miscalibrated imputation and propensity models, Kweon and Yu [26] propose a doubly calibrated estimator and a tri-level joint learning framework to simultaneously optimize calibration experts alongside prediction and imputation models. For the variance of estimators, an increasing body of works have emerged to reduce the variance. The most common estimator reducing variance is Self-Normalized IPS (SNIPS) [13]. Based on DR, literature [5] designed a MRDR estimator to reduce the variance of the DR estimator by the present variance expression of DR. In [15], TDR estimator is elaborated to reduce the bias and variance of DR simultaneously by the present semi-parametric collaborative learning. Moreover, stable DR estimator [12] achieves the bounded bias, variance, and generalization error bound simultaneously for arbitrarily small propensities by combining SNIPS and DR methods. Various regularization designs, such as SV [16], MIS [16], BMSE [8], and so forth, are also introduced into the estimator to achieve variance reduction.

# 6 Conclusions

To the best of our knowledge, this is the first work to reveal that the essence of estimator designs is not merely to eliminate bias, to reduce variance, or to achieve a simple bias-variance trade-off but to quantitatively and simultaneously optimize bias and variance. Besides, the limitations of general regularization techniques and general static estimators are presented. Based on the general laws with respect to the relationship between bias and variance, we propose a systematic dynamic learning framework, which guarantees the bounded variances and generalization bounds by the present fine-grained bias-variance joint optimization scheme. Extensive experiment results have verified the theoretical results and the performance of the present dynamic estimators. The search for optimal weights in the objective function and the functions in the dynamic estimation framework remains an open question.

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

## A  Derivation of bias and variance for naive, EIB, IPS, and DR estimators.

As mentioned in literature [11], the bias of an estimator is defined as

$$\text{Bias}(L) = |L_{\text{real}} - \mathbb{E}_O[L]| \tag{6}$$

According to the prediction inaccuracy expressions of $L_{\text{naive}}$ and the definition of bias (6), the bias of naive estimator satisfies

$$\text{Bias}(L_{\text{naive}}) = \frac{1}{|\mathcal{D}|}\left|\sum_{(u,i)\in\mathcal{D}} e_{u,i} - |\mathcal{D}|\mathbb{E}_O[L_{\text{naive}}]\right| = \frac{1}{|\mathcal{D}|}\left|\sum_{(u,i)\in\mathcal{D}}\left(1 - \frac{|\mathcal{D}|}{|\mathcal{O}|}p_{u,i}\right)e_{u,i}\right|. \tag{7}$$

According to the definition of variance for an estimation given in [5], the variance of the naive approach can be formulated as

$$\begin{aligned}
\mathbb{V}_O[L_{\text{naive}}] &= \mathbb{E}_O[L_{\text{naive}}^2] - \mathbb{E}_O^2[L_{\text{naive}}] \\
&= \frac{1}{|\mathcal{O}|^2}\mathbb{E}_O\left[\left(\sum_{(u,i)\in\mathcal{D}} o_{u,i}e_{u,i}\right)^2\right] - \frac{1}{|\mathcal{O}|^2}\left(\sum_{(u,i)\in\mathcal{D}} p_{u,i}e_{u,i}\right)^2 \\
&= \frac{1}{|\mathcal{O}|^2}\sum_{(u,i)\in\mathcal{D}} p_{u,i}(1 - p_{u,i})e_{u,i}^2
\end{aligned}$$

The biases of EIB, IPS and DR methods have been given in Lemma 3.1 of [11]. The variance formulations of IPS and DR estimators have been provided in [5]. For the variance of EIB, we obtain

$$\begin{aligned}
\mathbb{V}_O[L_{\text{EIB}}] &= \mathbb{E}_O[L_{\text{EIB}}^2] - \mathbb{E}_O^2[L_{\text{EIB}}] \\
&= \frac{1}{|\mathcal{D}|^2}\mathbb{E}_O\left[\left(\sum_{(u,i)\in\mathcal{D}}[o_{u,i}e_{u,i} + (1 - o_{u,i})\hat{e}_{u,i}]\right)^2\right] - \frac{1}{|\mathcal{D}|^2}\left(\sum_{(u,i)\in\mathcal{D}}[p_{u,i}e_{u,i} + (1 - p_{u,i})\hat{e}_{u,i}]\right)^2 \\
&= \frac{1}{|\mathcal{D}|^2}\sum_{(u,i)\in\mathcal{D}} p_{u,i}(1 - p_{u,i})(e_{u,i} - \hat{e}_{u,i})^2.
\end{aligned}$$

Therefore, the bias and variance of naive, EIB, IPS, and DR estimators shown in Table 4 can be obtained.

## B  Unbounded Variance

According to the definitions of variances of IPS and SR methods, if $\hat{p}_{u,i} = p_{u,i}$, when the propensity tends to zero, the variace of IPS and DR satisfy

$$\lim_{p_{u,i}\to 0}\mathbb{V}_O[L_{\text{IPS}}|\hat{p}_{u,i} = p_{u,i}] = \lim_{p_{u,i}\to 0}\frac{1}{|\mathcal{D}|^2}\sum_{(u,i)\in\mathcal{D}}\frac{1 - p_{u,i}}{p_{u,i}}e_{u,i}^2 = \infty,$$

$$\lim_{p_{u,i}\to 0}\mathbb{V}_O[L_{\text{DR}}|\hat{p}_{u,i} = p_{u,i}] = \lim_{p_{u,i}\to 0}\frac{1}{|\mathcal{D}|^2}\sum_{(u,i)\in\mathcal{D}}\frac{1 - p_{u,i}}{p_{u,i}}\delta_{u,i}^2 = \infty.$$

This demonstrates that the variances of IPS and DR are unbounded even if the propensities are accurate. If there exists one propensity going to zero then the variance tends to infinity

## C  Limitations of Regularization Techniques and Static Estimators

**Theorem 3.1.** Let $L_{\text{Est+Reg}}$ be defined in (1) and the estimator $L_{\text{Est}}$ be unbiased. If $L_{\text{Est+Reg}}$ is unbiased, then the variance of $L_{\text{Est+Reg}}$ is greater than the one of the original estimator $L_{\text{Est}}$.

Table 4: Bias and variance of naive, EIB, IPS, and DR estimators.

| Model | naive | EIB | IPS | DR |
|---|---|---|---|---|
| Bias | $\frac{1}{|\mathcal{O}|}\left|\sum_{(u,i)\in\mathcal{D}}\left(1 - p_{u,i}\right)e_{u,i}\right|$ | $\frac{1}{|\mathcal{D}|}\left|\sum_{(u,i)\in\mathcal{D}}(1 - p_{u,i})\delta_{u,i}\right|$ | $\frac{1}{|\mathcal{D}|}\left|\sum_{(u,i)\in\mathcal{D}}\Delta_{u,i}e_{u,i}\right|$ | $\frac{1}{|\mathcal{D}|}\left|\sum_{(u,i)\in\mathcal{D}}\Delta_{u,i}\delta_{u,i}\right|$ |
| Variance | $\frac{1}{|\mathcal{O}|^2}\sum_{(u,i)\in\mathcal{D}} p_{u,i}(1 - p_{u,i})e_{u,i}^2$ | $\frac{1}{|\mathcal{D}|^2}\sum_{(u,i)\in\mathcal{D}} p_{u,i}(1 - p_{u,i})\delta_{u,i}^2$ | $\frac{1}{|\mathcal{D}|^2}\sum_{(u,i)\in\mathcal{D}}\frac{p_{u,i}(1 - p_{u,i})}{\hat{p}_{u,i}^2}e_{u,i}^2$ | $\frac{1}{|\mathcal{D}|^2}\sum_{(u,i)\in\mathcal{D}}\frac{p_{u,i}(1 - p_{u,i})}{\hat{p}_{u,i}^2}\delta_{u,i}^2$ |

*Proof.* The variance of $L_{\text{Est+Reg}}$ satisfies

$$\mathbb{V}_O[L_{\text{Est+Reg}}] = \mathbb{V}_O[L_{\text{Est}}] + 2\lambda\text{Cov}(L_{\text{Est}}, L_{\text{Reg}}) + \lambda^2\mathbb{V}_O[L_{\text{Reg}}].$$

To reduce the variance of $L_{\text{Est}}$, $\mathbb{V}_O[L_{\text{Est+Reg}}]$ satisfies $\mathbb{V}_O[L_{\text{Est+Reg}}] \leq \mathbb{V}_O[L_{\text{Est}}]$, which implies that

$$2\lambda\text{Cov}(L_{\text{Est}}, L_{\text{Reg}}) + \lambda^2\mathbb{V}_O[L_{\text{Reg}}] \leq 0. \tag{8}$$

Therefore, the parameter $\lambda$ satisfies $0 \leq \lambda \leq -\frac{2\text{Cov}(L_{\text{Est}}, L_{\text{Reg}})}{\mathbb{V}_O[L_{\text{Reg}}]}$ and the optimal parameter is $\lambda_{\text{opt}} = -\frac{\text{Cov}(L_{\text{Est}}, L_{\text{Reg}})}{\mathbb{V}_O[L_{\text{Reg}}]}$. On the other hand, since $L_{\text{Est}}$ and $L_{\text{Est+Reg}}$ are unbiased, we obtain $\mathbb{E}_O[L_{\text{Reg}}] = 0$. Then $\text{Cov}(L_{\text{Est}}, L_{\text{Reg}})$ satisfies

$$
\begin{aligned}
\text{Cov}(L_{\text{Est}}, L_{\text{Reg}}) =& \mathbb{E}_O(L_{\text{Est}}L_{\text{Reg}}) - \mathbb{E}_O(L_{\text{Est}})\mathbb{E}_O(L_{\text{Reg}}) \\
=& \mathbb{E}_O(L_{\text{Est}}L_{\text{Reg}}) \\
=& \frac{1}{|\mathcal{D}|^2}\mathbb{E}_O\left(\left[\sum_{(u,i)\in\mathcal{D}}\left(f(o_{u,i}, \hat{p}_{u,i})e_{u,i} + g(o_{u,i}, \hat{p}_{u,i})\hat{e}_{u,i}\right)\right]\left[\sum_{(u,i)\in\mathcal{D}}h(o_{u,i}, \hat{p}_{u,i})\right]\right)
\end{aligned}
\tag{9}
$$

To facilitate representation, $f(o_{u,i}, \hat{p}_{u,i})e_{u,i} + g(o_{u,i}, \hat{p}_{u,i})\hat{e}_{u,i}$ is denoted as $r(o_{u,i}, \hat{p}_{u,i}, e_{u,i}, \hat{e}_{u,i})$, which satisfies $r(o_{u,i}, \hat{p}_{u,i}, e_{u,i}, \hat{e}_{u,i}) \geq 0$. Then the equation (9) can be rewritten as

$$
\begin{aligned}
\text{Cov}(L_{\text{Est}}, L_{\text{Reg}}) =& \frac{1}{|\mathcal{D}|^2}\mathbb{E}_O\left(\left[\sum_{(u,i)\in\mathcal{D}}r(o_{u,i}, \hat{p}_{u,i}, e_{u,i}, \hat{e}_{u,i})\right]\left[\sum_{(u,i)\in\mathcal{D}}h(o_{u,i}, \hat{p}_{u,i})\right]\right) \\
=& \frac{1}{|\mathcal{D}|^2}\mathbb{E}_O\left(\sum_{(u,i)\in\mathcal{D}}\left[h(o_{u,i}, \hat{p}_{u,i})\sum_{(u,i)\in\mathcal{D}}r(o_{u,i}, \hat{p}_{u,i}, e_{u,i}, \hat{e}_{u,i})\right]\right) \\
=& \frac{1}{|\mathcal{D}|^2}\mathbb{E}_O\left(\sum_{j=1}^{|D|}\sum_{k=1}^{|D|}h(o_j, \hat{p}_j)r(o_k, \hat{p}_k, e_k, \hat{e}_k)\right) \\
=& \frac{1}{|\mathcal{D}|^2}\sum_{j=1}^{|D|}\sum_{k=1}^{|D|}\mathbb{E}_O[h(o_j, \hat{p}_j)r(o_k, \hat{p}_k, e_k, \hat{e}_k)].
\end{aligned}
$$

Let us consider the term $\mathbb{E}_O[h(o_j, \hat{p}_j)r(o_k, \hat{p}_k, e_k, \hat{e}_k)]$ in (10), which fulfills $\mathbb{E}_O[h(o_j, \hat{p}_j)r(o_k, \hat{p}_k, e_k, \hat{e}_k)] \geq 0$. Therefore, we obtain $\text{Cov}(L_{Est}, L_{Reg}) = \mathbb{E}_O(L_{\text{Est}}L_{\text{Reg}}) \geq 0$. $\qquad\square$

**Corollary 3.2.** If the variance of $L_{\text{Est+Reg}}$ is less than the variance of the original estimator $L_{\text{Est}}$, then $L_{\text{Est+Reg}}$ is not unbiased.

*Proof.* We use the method of proof by contradiction. Assume that when $\mathbb{V}_O[L_{\text{Est+Reg}}] \leq \mathbb{V}_O[L_{\text{Est}}]$, $L_{\text{Est+Reg}}$ is unbiased. According to the definition of $\mathbb{V}_O[L_{\text{Est+Reg}}]$ and $\mathbb{V}_O[L_{\text{Est+Reg}}] \leq \mathbb{V}_O[L_{\text{Est}}]$, we have

$$
\begin{aligned}
\mathbb{V}_O[L_{\text{Est+Reg}}] =& \mathbb{V}_O[L_{\text{Est}}] + 2\lambda\text{Cov}(L_{\text{Est}}, L_{\text{Reg}}) + \lambda^2\mathbb{V}_O[L_{\text{Reg}}] \\
\leq& \mathbb{V}_O[L_{\text{Est}}],
\end{aligned}
\tag{10}
$$

which implies that $2\lambda\text{Cov}(L_{\text{Est}}, L_{\text{Reg}}) + \lambda^2\mathbb{V}_O[L_{\text{Reg}}] \leq 0$. Therefore, the parameter $\lambda$ needs to satisfy $0 \leq \lambda \leq -\frac{2\text{Cov}(L_{\text{Est}}, L_{\text{Reg}})}{\mathbb{V}_O[L_{\text{Reg}}]}$. Since $\mathbb{V}_O[L_{\text{Reg}}]$ and $\lambda \geq 0$, $\text{Cov}(L_{\text{Est}}, L_{\text{Reg}}) \leq 0$ needs to be satisfied. On the other hand, as shown in A1, when $L_{\text{Est+Reg}}$ is unbiased, we have $\text{Cov}(L_{Est}, L_{Reg}) = \mathbb{E}_O(L_{\text{Est}}L_{\text{Reg}}) \geq 0$, which contradicts the condition $\text{Cov}(L_{\text{Est}}, L_{\text{Reg}}) \leq 0$. Therefore, Corollary 3.2 holds. $\qquad\square$

**Theorem 3.3.** Let the bias of $L_{\text{Est+Reg}}$ be bounded and the variance of $L_{\text{Est}}$ satisfy $\lim_{p_{u,i}\to 0}\mathbb{V}_O[L_{\text{Est}}|\hat{p}_{u,i} = p_{u,i}] = \infty$. Then, there exists no regularizer $L_{\text{Reg}}$ that enables the variance and generalization bound of the estimator bounded even the learned imputed errors or propensities are accurate.

*Proof.* According to the definition of variance, $\mathbb{V}_O[L_{\text{Est+Reg}}]$ satisfies

$$
\begin{aligned}
\mathbb{V}_O[L_{\text{Est+Reg}}] =& \mathbb{E}_O[(L_{\text{Est}} + \lambda L_{\text{Reg}})^2] - \mathbb{E}_O^2[L_{\text{Est}} + \lambda L_{\text{Reg}}] \\
=& \mathbb{E}_O[L_{\text{Est}}^2 + \lambda^2 L_{\text{Reg}}^2 + 2\lambda L_{\text{Est}} L_{\text{Reg}}] - \mathbb{E}_O^2[L_{\text{Est}} + \lambda L_{\text{Reg}}].
\end{aligned}
\tag{11}
$$

Since the bias of the estimator $L_{\text{Est+Reg}}$ is bounded and $\lim_{p_{u,i}\to 0} \mathbb{V}_O[L_{\text{Est}}|\hat{p}_{u,i} = p_{u,i}] = \infty$, $\mathbb{E}_O^2[L_{\text{Est}} + \lambda L_{\text{Reg}}]$ is also bounded, i.e. $\mathbb{E}_O^2[L_{\text{Est}} + \lambda L_{\text{Reg}}] \leq \bar{B}$. Eq. (11) satisfies

$$
\begin{aligned}
\mathbb{V}_O[L_{\text{Est+Reg}}] =& \mathbb{E}_O[L_{\text{Est}}^2] + \lambda^2 \mathbb{E}_O[L_{\text{Reg}}^2] + 2\lambda \mathbb{E}_O[L_{\text{Est}} L_{\text{Reg}}] - \mathbb{E}_O^2[L_{\text{Est}} + \lambda L_{\text{Reg}}] \\
\geq& \mathbb{E}_O[L_{\text{Est}}^2] + \lambda^2 \mathbb{E}_O[L_{\text{Reg}}^2] + 2\lambda \mathbb{E}_O[L_{\text{Est}} L_{\text{Reg}}] - \bar{B}^2,
\end{aligned}
$$

which implies that $\lim_{p_{u,i}\to 0} \mathbb{V}_O[L_{\text{Est+Reg}}|\hat{p}_{u,i} = p_{u,i}] = \infty$. $\qquad\square$

**Theorem 3.4.** (Limitation of Static Estimator). Given prediction errors $e_{u,i}$, imputed errors $\hat{e}_{u,i}$, and learned propensities $\hat{p}_{u,i}$ for all user-item pairs $(u, i)$, if for any $e_{u,i} - g(0, \hat{p}_{u,i})\hat{e}_{u,i} \neq 0$, $L_{\text{Est}}$ given in (1) is unbiased, then the corresponding variance and generalization bound are unbounded.

*Proof.* According to the formulation of the estimator (1) and $f(0, \hat{p}_{u,i}) = 0$, its bias is given as

$$
\begin{aligned}
\text{Bias}(L) =& \frac{1}{|\mathcal{D}|} \left| \sum_{(u,i)\in\mathcal{D}} e_{u,i} - \mathbb{E}_O[L] \right| \\
=& \frac{1}{|\mathcal{D}|} \left| \sum_{(u,i)\in\mathcal{D}} \left[ (1 - f(1, \hat{p}_{u,i})p_{u,i})e_{u,i} - (g(1, \hat{p}_{u,i})p_{u,i} + g(0, \hat{p}_{u,i})(1 - p_{u,i}))\hat{e}_{u,i} \right] \right|.
\end{aligned}
\tag{12}
$$

From (12), it can be observed that the unbiasedness of the estimator $L$ implies that

$$
\left[1 - f(1, \hat{p}_{u,i})p_{u,i}\right]e_{u,i} - \left[g(1, \hat{p}_{u,i})p_{u,i} + g(0, \hat{p}_{u,i})(1 - p_{u,i})\right]\hat{e}_{u,i} = 0,
$$

which is equivalent to

$$
f(1, \hat{p}_{u,i})p_{u,i}e_{u,i} + \left[g(1, \hat{p}_{u,i})p_{u,i} + g(0, \hat{p}_{u,i})(1 - p_{u,i})\right]\hat{e}_{u,i} = e_{u,i}
\tag{13}
$$

and

$$
f(1, \hat{p}_{u,i})e_{u,i} + g(1, \hat{p}_{u,i})\hat{e}_{u,i} = \frac{e_{u,i} - g(0, \hat{p}_{u,i})(1 - p_{u,i})\hat{e}_{u,i}}{p_{u,i}}
\tag{14}
$$

Let us consider the variance of the estimator $L$. It satisfies

$$
\begin{aligned}
\mathbb{V}_O[L] =& \frac{1}{|\mathcal{D}|^2} \mathbb{E}_O\left[ \left( \sum_{(u,i)\in\mathcal{D}} \left( f(o_{u,i}, \hat{p}_{u,i})e_{u,i} + g(o_{u,i}, \hat{p}_{u,i})\hat{e}_{u,i} \right) \right)^2 \right] \\
& - \frac{1}{|\mathcal{D}|^2} \left( \sum_{(u,i)\in\mathcal{D}} \left[ f(1, \hat{p}_{u,i})p_{u,i}e_{u,i} + (g(1, \hat{p}_{u,i})p_{u,i} + g(0, \hat{p}_{u,i})(1 - p_{u,i}))\hat{e}_{u,i} \right] \right)^2 \\
=& \frac{1}{|\mathcal{D}|^2} \sum_{(u,i)\in\mathcal{D}} \left[ \mathbb{E}_O\left[ \left( f(o_{u,i}, \hat{p}_{u,i})e_{u,i} + g(o_{u,i}, \hat{p}_{u,i})\hat{e}_{u,i} \right)^2 \right] \right. \\
& \left. - \left[ f(1, \hat{p}_{u,i})p_{u,i}e_{u,i} + \left[g(1, \hat{p}_{u,i})p_{u,i} + g(0, \hat{p}_{u,i})(1 - p_{u,i})\right]\hat{e}_{u,i} \right]^2 \right] \\
=& \frac{1}{|\mathcal{D}|^2} \sum_{(u,i)\in\mathcal{D}} \left[ \mathbb{E}_O\left[ \left( f(o_{u,i}, \hat{p}_{u,i})e_{u,i} + g(o_{u,i}, \hat{p}_{u,i})\hat{e}_{u,i} \right)^2 \right] - e_{u,i}^2 \right] \qquad \text{because of (13).}
\end{aligned}
\tag{15}
$$

Let us focus on the fist term in (15). Denote $f(o_{u,i}, \hat{p}_{u,i})e_{u,i} + g(o_{u,i}, \hat{p}_{u,i})\hat{e}_{u,i}$ as $r(o_{u,i}, \hat{p}_{u,i}, e_{u,i}, \hat{e}_{u,i})$. Then, the first term in (15) satisfies

$$
\begin{aligned}
& \frac{1}{|\mathcal{D}|^2} \mathbb{E}_O\left[ \left( \sum_{(u,i)\in\mathcal{D}} r(o_{u,i}, \hat{p}_{u,i}, e_{u,i}, \hat{e}_{u,i}) \right)^2 \right] \\
& = \frac{1}{|\mathcal{D}|^2} \sum_{j=1}^{|\mathcal{D}|} \sum_{k=1}^{|\mathcal{D}|} \mathbb{E}_O\left[ r(o_j, \hat{p}_j, e_j, \hat{e}_j) r(o_k, \hat{p}_k, e_k, \hat{e}_k) \right] \\
& \geq \frac{1}{|\mathcal{D}|^2} \sum_{j=1}^{|\mathcal{D}|} \mathbb{E}_O\left[ r^2(o_j, \hat{p}_j, e_j, \hat{e}_j) \right] \qquad \text{because of } r(o_{u,i}, \hat{p}_{u,i}, e_{u,i}, \hat{e}_{u,i}) \geq 0 \\
& = \frac{1}{|\mathcal{D}|^2} \sum_{(u,i)\in\mathcal{D}} \mathbb{E}_O\left[ \left( f(o_{u,i}, \hat{p}_{u,i})e_{u,i} + g(o_{u,i}, \hat{p}_{u,i})\hat{e}_{u,i} \right)^2 \right]
\end{aligned}
\tag{16}
$$

According to (15) and (16), the variance of the estimator $L$ satisfies

$$
\begin{aligned}
\mathbb{V}_O[L] \geq & \frac{1}{|\mathcal{D}|^2} \sum_{(u,i)\in\mathcal{D}} \left[ \mathbb{E}_O\left[ \left( f(o_{u,i},\hat{p}_{u,i})e_{u,i} + g(o_{u,i},\hat{p}_{u,i})\hat{e}_{u,i} \right)^2 \right] \right. \\
& \left. - \left[ f(1,\hat{p}_{u,i})p_{u,i}e_{u,i} + \left[ g(1,\hat{p}_{u,i})p_{u,i} + g(0,\hat{p}_{u,i})(1-p_{u,i}) \right]\hat{e}_{u,i} \right]^2 \right] \\
= & \frac{1}{|\mathcal{D}|^2} \sum_{(u,i)\in\mathcal{D}} \left[ \mathbb{E}_O\left[ \left( f(o_{u,i},\hat{p}_{u,i})e_{u,i} + g(o_{u,i},\hat{p}_{u,i})\hat{e}_{u,i} \right)^2 \right] - e_{u,i}^2 \right] \\
= & \frac{1}{|\mathcal{D}|^2} \sum_{(u,i)\in\mathcal{D}} \left[ \left[ f(1,\hat{p}_{u,i})e_{u,i} + g(1,\hat{p}_{u,i})\hat{e}_{u,i} \right]^2 p_{u,i} + g^2(0,\hat{p}_{u,i})\hat{e}_{u,i}^2(1-p_{u,i}) - e_{u,i}^2 \right] \\
= & \frac{1}{|\mathcal{D}|^2} \sum_{(u,i)\in\mathcal{D}} \frac{1-p_{u,i}}{p_{u,i}} \left[ e_{u,i} - g(0,\hat{p}_{u,i})\hat{e}_{u,i} \right]^2 .
\end{aligned}
\tag{17}
$$

Therefore, when the propensity tends to zero, for any $e_{u,i} - g(0,\hat{p}_{u,i})\hat{e}_{u,i} \neq 0$, the limit of the variance satisfies $\lim_{p_{u,i}\to 0} \mathbb{V}_O[L] = \infty$. Since the generalization bound contains the bias and variance terms, the generalization bound is also unbounded when the propensity tends to zero. The proof is completed. $\square$

## D  Proofs of Properties for Fine-Grained Dynamic Estimators

**Lemma D.1** (Bias of D-IPS and D-DR). *Given prediction errors $e_{u,i}$, imputed errors $\hat{e}_{u,i}$, and learned propensities $\hat{p}_{u,i}$ for all user-item pairs $(u,i)$, the biases of the D-IPS and D-DR methods are given as*

$$
Bias(L_{\text{D-IPS}}) = \frac{1}{|\mathcal{D}|} \left| \sum_{(u,i)\in\mathcal{D}} h_B^{Est}(\hat{p}_{u,i},p_{u,i},\alpha)e_{u,i} \right|, \; Bias(L_{\text{D-DR}}) = \frac{1}{|\mathcal{D}|} \left| \sum_{(u,i)\in\mathcal{D}} h_B^{Est}(\hat{p}_{u,i},p_{u,i},\alpha)\delta_{u,i} \right|,
\tag{18}
$$

*where the function $h_B$ satisfies $h_B^{Est}(\hat{p}_{u,i},p_{u,i},\alpha) = 1 - \frac{p_{u,i}}{f^\alpha(\hat{p}_{u,i})}$.*

*Proof.* According to the definition of bias (6), the biases of D-IPS and D-DR are formulated as

$$
\begin{aligned}
\text{Bias}(L_{\text{D-IPS}}) = & \frac{1}{|\mathcal{D}|} \left| \sum_{(u,i)\in\mathcal{D}} e_{u,i} - \mathbb{E}_O[L_{\text{D-IPS}}] \right| = \frac{1}{|\mathcal{D}|} \left| \sum_{(u,i)\in\mathcal{D}} \left( 1 - \frac{p_{u,i}}{f^\alpha(\hat{p}_{u,i})} \right)e_{u,i} \right|, \\
\text{Bias}(L_{\text{D-DR}}) = & \frac{1}{|\mathcal{D}|} \left| \sum_{(u,i)\in\mathcal{D}} e_{u,i} - \mathbb{E}_O[L_{\text{D-DR}}] \right| = \frac{1}{|\mathcal{D}|} \left| \sum_{(u,i)\in\mathcal{D}} \left( 1 - \frac{p_{u,i}}{f^\alpha(\hat{p}_{u,i})} \right)\delta_{u,i} \right|.
\end{aligned}
\tag{19}
$$

$\square$

**Lemma D.2** (Variance of D-IPS and D-DR). *Given $e_{u,i}$, $\hat{e}_{u,i}$, and $\hat{p}_{u,i}$ for all $(u,i) \in \mathcal{D}$, the variances of the D-IPS and D-DR methods are given as*

$$
\mathbb{V}_O[L_{\text{D-IPS}}] = \frac{1}{|\mathcal{D}|^2} \sum_{(u,i)\in\mathcal{D}} h_V^{Est}(\hat{p}_{u,i},p_{u,i},\alpha)e_{u,i}^2, \; \mathbb{V}_O[L_{\text{D-DR}}] = \frac{1}{|\mathcal{D}|^2} \sum_{(u,i)\in\mathcal{D}} h_V^{Est}(\hat{p}_{u,i},p_{u,i},\alpha)\delta_{u,i}^2,
$$

*where the function $h_V$ satisfies $h_V^{Est}(\hat{p}_{u,i},p_{u,i},\alpha) = \frac{p_{u,i}(1-p_{u,i})}{f^{2\alpha}(\hat{p}_{u,i})}$.*

*Proof.* Considering the definition of the variance, we obtain the variances of D-IPS and D-DR as

$$
\begin{aligned}
\mathbb{V}_O[L_{\text{D-IPS}}] = & \mathbb{E}_O[L_{\text{D-IPS}}^2] - \mathbb{E}_O^2[L_{\text{D-IPS}}] \\
= & \frac{1}{|\mathcal{D}|^2} \mathbb{E}_O\left[ \left( \sum_{(u,i)\in\mathcal{D}} \frac{o_{u,i}}{f^\alpha(\hat{p}_{u,i})}e_{u,i} \right)^2 \right] - \frac{1}{|\mathcal{D}|^2} \left( \sum_{(u,i)\in\mathcal{D}} \frac{p_{u,i}}{f^\alpha(\hat{p}_{u,i})}e_{u,i} \right)^2 \\
= & \frac{1}{|\mathcal{D}|^2} \sum_{(u,i)\in\mathcal{D}} \frac{p_{u,i}(1-p_{u,i})}{f^{2\alpha}(\hat{p}_{u,i})}e_{u,i}^2
\end{aligned}
$$

and

$$\mathbb{V}_O[L_{\text{D-DR}}] = \mathbb{E}_O[L_{\text{D-DR}}^2] - \mathbb{E}_O^2[L_{\text{EIB}}]$$

$$= \frac{1}{|\mathcal{D}|^2}\mathbb{E}_O\bigg[\bigg(\sum_{(u,i)\in\mathcal{D}}\frac{o_{u,i}}{f^\alpha(\hat{p}_{u,i})}\delta_{u,i}\bigg)^2\bigg] - \frac{1}{|\mathcal{D}|^2}\bigg(\sum_{(u,i)\in\mathcal{D}}\frac{p_{u,i}}{f^\alpha(\hat{p}_{u,i})}\delta_{u,i}\bigg)^2$$

$$= \frac{1}{|\mathcal{D}|^2}\sum_{(u,i)\in\mathcal{D}}\frac{p_{u,i}(1-p_{u,i})}{f^{2\alpha}(\hat{p}_{u,i})}\delta_{u,i}^2.$$

$\square$

**Proposition D.3** (Monotonicity of Bias and Variance). *For IPS-based and DR-based dynamic learning frameworks, and given $e_{u,i}, \hat{e}_{u,i}, \hat{p}_{u,i}$ for all $(u,i) \in \mathcal{D}$, if $f(\hat{p}_{u,i}) \geq \hat{p}_{u,i}$ and the parameter $\alpha \in [0,1]$ is increasing, then biases of D-IPS and D-DR are monotonically decreasing and their variances are monotonically increasing when learned propensities are accurate.*

*Proof.* According to the determining functions $h_B(\hat{p}_{u,i}, p_{u,i}, \alpha)$ and $h_V(\hat{p}_{u,i}, p_{u,i}, \alpha)$ in the bias and variance, the first derivative of $h_B$ and $h_V$ versus $\alpha$ are derived as

$$\frac{\partial h_B(\hat{p}_{u,i}, p_{u,i}, \alpha)}{\partial\alpha} = \frac{p_{u,i}\ln(f(\hat{p}_{u,i}))}{f^\alpha(\hat{p}_{u,i})}, \quad \frac{\partial h_V(\hat{p}_{u,i}, p_{u,i}, \alpha)}{\partial\alpha} = -\frac{2p_{u,i}(1-p_{u,i})\ln(f(\hat{p}_{u,i}))}{f^{2\alpha}(\hat{p}_{u,i})}. \quad (20)$$

For all $\hat{p}_{u,i} \in (0,1)$, we have $\ln(f(\hat{p}_{u,i})) < 0$. Therefore, $\frac{\partial h_B(\hat{p}_{u,i}, p_{u,i}, \alpha)}{\partial\alpha} < 0$ and $\frac{\partial h_V(\hat{p}_{u,i}, p_{u,i}, \alpha)}{\partial\alpha} > 0$ result in the monotonically decreasing function $h_B$ and the monotonically increasing $h_V$ for all user-item pairs $(u,i)$ as the number of $\alpha \in [0,1]$ increases. Note the function $f(\hat{p}_{u,i})$ satisfies $f(\hat{p}_{u,i}) \geq \hat{p}_{u,i}$, which implies that $h_B \geq 0$ and $h_V > 0$ when learned propensities are accurate. Therefore, biases of D-IPS and D-DR are monotonically decreasing and their variances are monotonically increasing when learned propensities are accurate. $\square$

**Lemma D.4** (Tail Bounds of D-IPS and D-DR). *Given $\hat{e}_{u,i}$, and $\hat{p}_{u,i}$ for all $(u,i) \in \mathcal{D}$, for any prediction results, with probability $1 - \rho$, the deviation of D-IPS and D-DR estimators from their expectations satisfy*

$$\Big|L_{\text{D-IPS}} - \mathbb{E}_O[L_{\text{D-IPS}}]\Big| \leq \sqrt{\frac{\log(\frac{2}{\rho})}{2|\mathcal{D}|^2}\sum_{(u,i)\in\mathcal{D}}\bigg(\frac{e_{u,i}}{f^\alpha(\hat{p}_{u,i})}\bigg)^2},$$

$$\Big|L_{\text{D-DR}} - \mathbb{E}_O[L_{\text{D-DR}}]\Big| \leq \sqrt{\frac{\log(\frac{2}{\rho})}{2|\mathcal{D}|}\sum_{(u,i)\in\mathcal{D}}\bigg(\frac{\delta_{u,i}}{f^\alpha(\hat{p}_{u,i})}\bigg)^2}.$$

$(21)$

*Proof.* Let $X_{u,i}^{\text{D-IPS}}$ and $X_{u,i}^{\text{D-DR}}$ be new random variables, which are defined as $X_{u,i}^{\text{D-IPS}} = \frac{o_{u,i}}{f^\alpha(\hat{p}_{u,i})}e_{u,i}$ and $X_{u,i}^{\text{D-DR}} = \hat{e}_{u,i} + \frac{o_{u,i}}{f^\alpha(\hat{p}_{u,i})}\delta_{u,i}$, respectively. Considering the independent observation indicators $\{o_{u,i}|(u,i)\in\mathcal{D}\}$, random variables $\{X_{u,i}^{\text{D-IPS}}|(u,i)\in\mathcal{D}\}$ and $\{X_{u,i}^{\text{D-DR}}|(u,i)\in\mathcal{D}\}$ are independent of each other. Then, the probability distributions of $X_{u,i}^{\text{D-IPS}}$ and $X_{u,i}^{\text{D-DR}}$ can be obtained as follows:

$$\Pr\bigg(X_{u,i}^{\text{D-IPS}} = \frac{e_{u,i}}{f^\alpha(\hat{p}_{u,i})}\bigg) = p_{u,i}, \; \Pr(X_{u,i}^{\text{D-IPS}} = 0) = 1 - p_{u,i},$$

$$\Pr\bigg(X_{u,i}^{\text{D-DR}} = \hat{e}_{u,i} + \frac{\delta_{u,i}}{f^\alpha(\hat{p}_{u,i})}\bigg) = p_{u,i}, \; \Pr(X_{u,i}^{\text{D-DR}} = \hat{e}_{u,i}) = 1 - p_{u,i}$$

According to the Hoeffding's inequality, for any $\varepsilon > 0$, we have the following inequality

$$\Pr\bigg(\Big|\sum_{(u,i)\in\mathcal{D}}X_{u,i} - \mathbb{E}_O\Big[\sum_{(u,i)\in\mathcal{D}}X_{u,i}\Big]\Big| \geq \varepsilon\bigg) \leq 2\exp\bigg(\frac{-2\varepsilon^2}{\sum_{(u,i)\in\mathcal{D}}g^2(\hat{p}_{u,i}, z_{u,i})}\bigg), \quad (22)$$

where $g(\hat{p}_{u,i}, e_{u,i}) = \frac{e_{u,i}}{f^\alpha(\hat{p}_{u,i})}$ for D-IPS and $g(\hat{p}_{u,i}, \delta_{u,i}) = \frac{\delta_{u,i}}{f^\alpha(\hat{p}_{u,i})}$ for D-DR. Let $\gamma = \frac{\varepsilon}{|\mathcal{D}|}$. Therefore, (22) can be rewritten as

$$\Pr\bigg(\Big|L_{\text{D-IPS}} - \mathbb{E}_O[L_{\text{D-IPS}}]\Big| \geq \gamma\bigg) \leq 2\exp\bigg(\frac{-2(\gamma|\mathcal{D}|)^2}{\sum_{(u,i)\in\mathcal{D}}g^2(\hat{p}_{u,i}, z_{u,i})}\bigg). \quad (23)$$

Let $\Pr\bigg(\Big|L_{\text{D-IPS}} - \mathbb{E}_O[L_{\text{D-IPS}}]\Big| \geq \gamma\bigg) = \rho$. According to the inequality (23), the errors $\gamma$ for D-IPS and D-DR can be solved as in (21). $\square$

**Theorem 3.5.** (The optimal parameter $\alpha_{u,i}^{\text{opt}}$). Let the learned propensities be accurate, i.e., $\hat{p}_{u,i} = p_{u,i}$. For weights $w_1$ and $w_2$, the objective function $w_1 h_B^{\text{Est}} + w_2 h_V^{\text{Est}}$ under $\alpha \in [0,1]$ achieves its minimum at

$$\alpha_{\text{opt}} = \min \left\{ \max \left\{ \frac{\ln\left(\frac{2w_2}{w_1}(1 - p_{u,i})\right)}{\ln(f(p_{u,i}))}, 0 \right\}, 1 \right\}. \tag{24}$$

*Proof.* The first derivative of the objective function $w_1 h_B^{\text{Est}}(\alpha_{\text{opt}}) + w_2 h_V^{\text{Est}}(\alpha_{\text{opt}})$ versus $\alpha$ is derived as

$$
\begin{aligned}
\frac{\partial \text{Objective}(\hat{p}_{u,i}, p_{u,i}, \alpha)}{\partial \alpha} &= w_1 \frac{\partial h_B^{\text{Est}}(\hat{p}_{u,i}, p_{u,i}, \alpha)}{\partial \alpha} + w_2 \frac{\partial h_B^{\text{Est}}(\hat{p}_{u,i}, p_{u,i}, \alpha)}{\partial \alpha} \\
&= w_1 \frac{p_{u,i} \ln(f(\hat{p}_{u,i}))}{f^{\alpha}(\hat{p}_{u,i})} - w_2 \frac{2p_{u,i}(1 - p_{u,i}) \ln(f(\hat{p}_{u,i}))}{f^{2\alpha}(\hat{p}_{u,i})}.
\end{aligned} \tag{25}
$$

Let $\frac{\partial \text{Objective}(\alpha | \hat{p}_{u,i} = p_{u,i})}{\partial \alpha}$ be zero. Then the optimal $\alpha$ satisfies

$$\alpha_{\text{opt}} = \frac{\ln\left(\frac{2w_2}{w_1}(1 - p_{u,i})\right)}{\ln(f(p_{u,i}))}. \tag{26}$$

Note that $\alpha$ needs to fulfill $0 \le \alpha \le 1$. Therefore, the solution of the optimization problem with the constraint can be obtained. □

**Theorem 3.6.** (Generalization Bounds of D-IPS and D-DR). For any finite hypothesis space $\mathcal{H}$ of $\hat{Y}$ and the optimal prediction matrix $\hat{Y}^-$, given $\hat{e}_{u,i}$ and $\hat{p}_{u,i}$ for all $(u,i) \in \mathcal{D}$, with probability $1 - \rho$, the prediction inaccuracies $L_{\text{D-IPS}}(\hat{Y}^-, Y)$ and $L_{\text{D-DR}}(\hat{Y}^-, Y)$ under D-IPS and D-DR have the following upper bounds

$$L_{\text{D-IPS}}(\hat{Y}^-, Y^O) + \sum_{(u,i) \in \mathcal{D}} \frac{|h_B^{\text{Est}}(\hat{p}_{u,i}, p_{u,i}, \alpha) e_{u,i}^-|}{|\mathcal{D}|} + h_G^{\text{Est}}(e_{u,i}^+),$$

$$L_{\text{D-DR}}(\hat{Y}^-, Y^O) + \sum_{(u,i) \in \mathcal{D}} \frac{|h_B^{\text{Est}}(\hat{p}_{u,i}, p_{u,i}, \alpha) \delta_{u,i}^-|}{|\mathcal{D}|} + h_G^{\text{Est}}(\delta_{u,i}^+),$$

where $e_{u,i}^+$ and $\delta_{u,i}^+$ are the error and error deviation corresponding to $\hat{Y}^+ = \arg\max_{\hat{Y} \in \mathcal{H}} \left\{ \sum_{(u,i) \in \mathcal{D}} \left(\frac{e_{u,i}}{f^{\alpha}(\hat{p}_{u,i})}\right)^2 \right\}$ and $\hat{Y}^+ = \arg\max_{\hat{Y} \in \mathcal{H}} \left\{ \sum_{(u,i) \in \mathcal{D}} \left(\frac{\delta_{u,i}}{f^{\alpha}(\hat{p}_{u,i})}\right)^2 \right\}$, respectively, and the function $h_G^{\text{Est}}$ is formulated as

$$h_G^{\text{Est}}(z_{u,i}^+) = \sqrt{\frac{\log(\frac{2|\mathcal{H}|}{\rho})}{2|\mathcal{D}|^2} \sum_{(u,i) \in \mathcal{D}} \left(\frac{z_{u,i}^+}{f^{\alpha}(\hat{p}_{u,i})}\right)^2} \tag{27}$$

*Proof.* According to the definition of bias, the differences between $L_{\text{real}}(\hat{Y}, Y)$ and expectations of $L_{\text{D-IPS}}(\hat{Y}^-, Y^O)$ and $L_{\text{D-DR}}(\hat{Y}^-, Y^O)$ satisfy

$$
\begin{aligned}
L_{\text{real}}(\hat{Y}^-, Y) - L_{\text{D-IPS}}(\hat{Y}^-) &= L_{\text{real}}(\hat{Y}^-, Y) - \mathbb{E}_O[L_{\text{D-IPS}}(\hat{Y}^-)] + \mathbb{E}_O[L_{\text{D-IPS}}(\hat{Y}^-)] - L_{\text{D-IPS}}(\hat{Y}^-) \\
&\le \text{Bias}(L_{\text{D-IPS}}(\hat{Y}^-)) + \mathbb{E}_O[L_{\text{D-IPS}}(\hat{Y}^-)] - L_{\text{D-IPS}}(\hat{Y}^-)
\end{aligned} \tag{28}
$$

and

$$
\begin{aligned}
L_{\text{real}}(\hat{Y}^-, Y) - L_{\text{D-DR}}(\hat{Y}^-) &= L_{\text{real}}(\hat{Y}^-, Y) - \mathbb{E}_O[L_{\text{D-DR}}(\hat{Y}^-)] + \mathbb{E}_O[L_{\text{D-DR}}(\hat{Y}^-)] - L_{\text{D-DR}}(\hat{Y}^-) \\
&\le \text{Bias}(L_{\text{D-DR}}(\hat{Y}^-)) + \mathbb{E}_O[L_{\text{D-DR}}(\hat{Y}^-)] - L_{\text{D-DR}}(\hat{Y}^-),
\end{aligned} \tag{29}
$$

respectively. From the inequalities (28) and (29), the expressions of $\text{Bias}(L_{\text{D-IPS}}(\hat{Y}^-))$ and $\text{Bias}(L_{\text{D-DR}}(\hat{Y}^-))$ have been given in Lemma 3.6 and, in what follows, terms $\mathbb{E}_O[L_{\text{D-IPS}}(\hat{Y}^-)] - L_{\text{D-IPS}}(\hat{Y}^-)$ in (28) and $\mathbb{E}_O[L_{\text{D-IPS}}(\hat{Y}^-)] - L_{\text{D-IPS}}(\hat{Y}^-)$ in (29) are discussed. Considering the finite

hypothesis space $\mathcal{H} = \{\hat{Y}^1, \hat{Y}^2, \ldots, \hat{Y}^{|\mathcal{H}|}\}$ and the Hoeffding's inequality, for any $\varepsilon > 0$, the following inequalities can be obtained

$$
\begin{aligned}
\Pr\Big(\Big|L_{\text{D-IPS}}(\hat{Y}^-) - \mathbb{E}_O[L_{\text{D-IPS}}(\hat{Y}^-)]\Big| \leq \gamma\Big) &= 1 - \Pr\Big(\Big|L_{\text{D-IPS}}(\hat{Y}^-) - \mathbb{E}_O[L_{\text{D-IPS}}(\hat{Y}^-)]\Big| \geq \gamma\Big) \\
&\geq 1 - \Pr\Big(\max_{\hat{Y}^\ell \in \mathcal{H}} \Big|L_{\text{D-IPS}}(\hat{Y}^\ell) - \mathbb{E}_O[L_{\text{D-IPS}}(\hat{Y}^\ell)]\Big| \geq \gamma\Big) \\
&\geq 1 - \sum_{\ell=1}^{\mathcal{H}} \Pr\Big(\Big|L_{\text{D-IPS}}(\hat{Y}^\ell) - \mathbb{E}_O[L_{\text{D-IPS}}(\hat{Y}^\ell)]\Big| \geq \gamma\Big) \\
&= 1 - \sum_{\ell=1}^{\mathcal{H}} 2\exp\Big(\frac{-2(\gamma|\mathcal{D}|)^2}{\sum\limits_{(u,i)\in\mathcal{D}} g^2(\hat{p}_{u,i}, e_{u,i}^\ell)}\Big) \\
&\geq 1 - 2|\mathcal{H}|\exp\Big(\frac{-2(\gamma|\mathcal{D}|)^2}{\sum\limits_{(u,i)\in\mathcal{D}} g^2(\hat{p}_{u,i}, e_{u,i}^+)}\Big)
\end{aligned}
\tag{30}
$$

$$
\Pr\Big(\Big|L_{\text{D-DR}}(\hat{Y}^-) - \mathbb{E}_O[L_{\text{D-DR}}(\hat{Y}^-)]\Big| \leq \gamma\Big) \geq 1 - 2|\mathcal{H}|\exp\Big(\frac{-2(\gamma|\mathcal{D}|)^2}{\sum\limits_{(u,i)\in\mathcal{D}} g^2(\hat{p}_{u,i}, \delta_{u,i}^+)}\Big).
$$

Let $2|\mathcal{H}|\exp\Big(\frac{-2(\gamma|\mathcal{D}|)^2}{\sum\limits_{(u,i)\in\mathcal{D}} g^2(\hat{p}_{u,i}, Z_{u,i}^+)}\Big)$ be $\rho$. The errors $\gamma_{\text{D-IPS}}$ and $\gamma_{\text{D-DR}}$ for D-IPS and D-DR can be solved as

$$
\gamma_{\text{D-IPS}} = \sqrt{\frac{\log(\frac{2|\mathcal{H}|}{\rho})}{2|\mathcal{D}|^2} \sum_{(u,i)\in\mathcal{D}} \Big(\frac{e_{u,i}^+}{f^\alpha(\hat{p}_{u,i})}\Big)^2}, \quad \gamma_{\text{D-DR}} = \sqrt{\frac{\log(\frac{2|\mathcal{H}|}{\rho})}{2|\mathcal{D}|^2} \sum_{(u,i)\in\mathcal{D}} \Big(\frac{\delta_{u,i}^+}{f^\alpha(\hat{p}_{u,i})}\Big)^2}.
\tag{31}
$$

Therefore, $\mathbb{E}_O[L_{\text{D-IPS}}(\hat{Y}^-)] - L_{\text{D-IPS}}(\hat{Y}^-)$ in (28) and $\mathbb{E}_O[L_{\text{D-IPS}}(\hat{Y}^-)] - L_{\text{D-IPS}}(\hat{Y}^-)$ in (29) fulfill

$$
\begin{aligned}
\mathbb{E}_O[L_{\text{D-IPS}}(\hat{Y}^-)] - L_{\text{D-IPS}}(\hat{Y}^-) &\leq \sqrt{\frac{\log(\frac{2|\mathcal{H}|}{\rho})}{2|\mathcal{D}|^2} \sum_{(u,i)\in\mathcal{D}} \Big(\frac{e_{u,i}^+}{f^\alpha(\hat{p}_{u,i})}\Big)^2}, \\
\mathbb{E}_O[L_{\text{D-DR}}(\hat{Y}^-)] - L_{\text{D-DR}}(\hat{Y}^-) &\leq \sqrt{\frac{\log(\frac{2|\mathcal{H}|}{\rho})}{2|\mathcal{D}|^2} \sum_{(u,i)\in\mathcal{D}} \Big(\frac{\delta_{u,i}^+}{f^\alpha(\hat{p}_{u,i})}\Big)^2}.
\end{aligned}
\tag{32}
$$

Combining (28), (29) and (32), we can obtain the generalization bounds of D-IPS and D-DR given in Lemma 3.11. $\qquad\square$

**Theorem 3.7.** (Boundedness of Variance and Generalization Bounds). Let $\alpha_{u,i}^{\text{opt}} \in [0,1]$ be the optimal parameter of (4). If the dynamic estimators adopt $\alpha_{u,i}^{\text{opt}}$ as the parameter, then the corresponding variance and generalization bounds are bounded.

*Proof.* Considering the optimal parameter $\alpha_{u,i}^{\text{opt}}$ for each user-item pair $(u,i)$ and the optimization problem (4), we can obtain the corresponding optimal objective function

$$
\text{Objective}^{\text{opt}} = w_1 E_B(h_B^{\text{Est}}(\alpha_{u,i}^{\text{opt}})) + w_2 E_V(h_V^{\text{Est}}(\alpha_{u,i}^{\text{opt}})) \leq w_1 E_B(h_B^{\text{Est}}(0)) + w_2 E_V(h_V^{\text{Est}}(0)).
\tag{33}
$$

Since $h_B^{\text{Est}}(\hat{p}_{u,i}, p_{u,i}, \alpha) > 0$ and $h_V^{\text{Est}}(\hat{p}_{u,i}, p_{u,i}, \alpha) > 0$, considering (33), we have

$$
\begin{aligned}
w_2 E_V(h_V^{\text{Est}}(\alpha_{u,i}^{\text{opt}})) &\leq w_1 E_B(h_B^{\text{Est}}(0)) + w_2 E_V(h_V^{\text{Est}}(0)) \\
&= w_1 E_B(1 - p_{u,i}) + w_2 E_V(p_{u,i}(1 - p_{u,i})),
\end{aligned}
\tag{34}
$$

which implies that

$$
\begin{aligned}
h_V^{\text{Est}}(\hat{p}_{u,i}, p_{u,i}, \alpha_{u,i}^{\text{opt}}) &= \frac{p_{u,i}(1 - p_{u,i})}{f^{2\alpha_{u,i}^{\text{opt}}}(\hat{p}_{u,i})} \\
&\leq E_V^{-1}\Big(\frac{w_1 E_B(1 - p_{u,i})}{w_2} + E_V(p_{u,i}(1 - p_{u,i}))\Big) \\
&= E_V^{-1}\Big(\frac{w_1 E_B(1)}{w_2} + E_V(0.25)\Big).
\end{aligned}
\tag{35}
$$

Therefore, the variance of dynamic estimators are bounded by

$$
\begin{aligned}
\mathbb{V}_O[L_{\text{D-IPS}}|\alpha = \alpha_{u,i}^{\text{opt}}] =& \frac{1}{|\mathcal{D}|^2} \sum_{(u,i)\in\mathcal{D}} h_V^{\text{Est}}(\hat{p}_{u,i}, p_{u,i}, \alpha_{u,i}^{\text{opt}}) e_{u,i}^2 \\
\leq& \frac{1}{|\mathcal{D}|^2} \sum_{(u,i)\in\mathcal{D}} E_V^{-1}\Big(\frac{w_1 E_B(1)}{w_2} + E_V(0.25)\Big) e_{u,i}^2, \\
\mathbb{V}_O[L_{\text{D-DR}}|\alpha = \alpha_{u,i}^{\text{opt}}] =& \frac{1}{|\mathcal{D}|^2} \sum_{(u,i)\in\mathcal{D}} h_V^{\text{Est}}(\hat{p}_{u,i}, p_{u,i}, \alpha_{u,i}^{\text{opt}}) \delta_{u,i}^2 \\
\leq& \frac{1}{|\mathcal{D}|^2} \sum_{(u,i)\in\mathcal{D}} E_V^{-1}\Big(\frac{w_1 E_B(1)}{w_2} + E_V(0.25)\Big) \delta_{u,i}^2.
\end{aligned}
\tag{36}
$$

Considering the expression of $h_G^{\text{Est}}(z_{u,i}^+)$ in generalization bounds and the boundedness of $h_V^{\text{Est}}(\hat{p}_{u,i}, p_{u,i}, \alpha_{u,i}^{\text{opt}})$, it is easy to obtain that under $\rho \neq 0$ the generalization bounds of $L_{\text{D-IPS}}$ and $L_{\text{D-DR}}$ are bounded. $\qquad\square$

