# OpenReview forum: "Fine-Grained Dynamic Framework for Bias-Variance Joint Optimization on Data Missing Not at Random"
_NeurIPS.cc/2024/Conference — NeurIPS 2024 poster_

### Official Review · Reviewer_rrSW · 2024-06-15

**Soundness:** 2
**Presentation:** 3
**Contribution:** 3
**Rating:** 5
**Confidence:** 3

**Summary:**

This paper analyzes the relationship between bias, variance, and generalization bound of proposed general estimator with data missing not at random, and proposes a quantitative bias-variance joint optimization method to achieve bounded variance.

**Strengths:**

- S1: The research question is good, and investigating debiasing methods when data is biased can be beneficial in real-world applications.
- S2: The authors provide some theoretical insights and proofs.
- S3: The authors conduct statistical significance test on the experimental results.

**Weaknesses:**

- W1: The organization and expression of the article need improvement. The current layout of figures, tables, theorems, and formulas is somewhat crowded, which hinders readability.
- W2: The discussion on the limitations of previous methods is not accurately described, and some expressions lack precision. For example, in lines 99-100, the boundedness of the variances of IPS and DR estimators is not related to the accuracy of the estimated propensities; instead, it is related to whether there are extremely small values among these estimates. As shown in the Table 4, if the estimated propensities are accurate and there are no extremely small values, the variances of both IPS and DR are bounded.
- W3: The proof of Theorem 3.1 requires further discussion, and the validity of the final inequality in the proof at line 739 is questionable, as it depends on the specific forms of functions $f$, $g$, and $h$, which significantly influence the final conclusion. For instance, assuming $f_{1}$, $g_{1}$, and $h_{1}$ satisfy $cov(L_{Est}, L_{Reg}) \geq 0$ , then setting $h_{2} = -h_{1}$ results in $cov(L_{Est}, L_{Reg}) \leq 0$. Therefore, it remains to be discussed whether this conclusion holds for any arbitrary function. If I have misunderstood, please correct me.
- W4: The authors validate the effectiveness of their method using only two real-world datasets, which lacks sufficient persuasiveness. For experiments on real-world datasets, it is recommended that the authors further validate their findings using the KuaiRec dataset [1].
- W5: Typos, for example, 'is' in line 11 and 'click, conversion' in line 24.

**References**

[1] KuaiRec: A Fully-observed Dataset and Insights for Evaluating Recommender Systems. CIKM 2022.

**Questions:**

- Q1: In the proof of Theorem 3.4, the second equality at line 757 is not evidently clear and requires further elaboration. Specifically, I am uncertain whether the order of summation and squaring operations is commutative as suggested in the proof.
- Q2: The formal definitions of symbols $E_{B}$ and $E_{V}$ are absent, which are important in core Theorem 3.7. Furthermore, their associated properties such as reversibility and boundedness are under-discussed.
- Q3: The function $f^\alpha(\hat p_{u,i})$ is manually designed and selected, but the advantages of two design criteria Isotonic Propensity and Same Order, have not been discussed in theory or experimentation.

**Limitations:**

Please see the "Weakness" and "Questions".

---

> ### Author Rebuttal · Authors · 2024-08-07
>
> **Response for Weaknesses**
> A1. For the discussion of figures, the descriptions of theoretical results and their proofs, and experimants, according to comments from reviewers, we will add more explanation to avoid any ambiguity and improve the readability.
>
> A2. Sorry for the unprecise descriptions. We will modify the descriptions in lines 99-100 to the following descriptions.
>
> *''..., then IPS and DR estimators are unbiased. For a new dataset, we cannot know in advance the range of the propensities in this dataset. Therefore, a new dataset may introduce extremely small propensities to lead to unbounded variances of IPS and DR, which will disrupt the stability of estimators, especially for larger datasets. It is unacceptable for real industrial scenarios.''*
>
> A3. Considering (1), for all $(u,i)$, we have $f(o _{u,i},\hat{p} _{u,i})e _{u,i}+g(o _{u,i},\hat{p} _{u,i})\hat{e} _{u,i}\ge0$ and $h(o _{u,i},\hat{p} _{u,i})\ge0$. Let $f(o _{u,i},\hat{p} _{u,i})e _{u,i}+g(o _{u,i},\hat{p} _{u,i})\hat{e} _{u,i}$ be denoted as $r(o _{u,i},\hat{p} _{u,i},e _{u,i},\hat{e} _{u,i})$. As given in the proof of Theorem 3.1, we have
> $$\begin{align}\text{Cov}(L _\text{Est}, L _\text{Reg})\=\frac{1}{\vert\mathcal{D}\vert^2}\sum _{j=1}^{\vert{D}\vert}\sum _{k=1}^{\vert{D}\vert}\mathbb{E} _O[h(o _j,\hat{p} _j)r(o _k,\hat{p} _k,e _k,\hat{e} _k)].\end{align}$$
> Since $\mathbb{E} _O[h(o _j,\hat{p} _j)r(o _k,\hat{p} _k,e _k,\hat{e} _k)]\ge0$, we obtain $\text{Cov}(L _{Est}, L _{Reg})=\mathbb{E} _O(L _\text{Est}L _\text{Reg})\ge0$.
>
> We will add the above formula to proof of Theorem 3.1.
>
> A4. The baseline approaches and the dynamic estimators are conducted on the KuaiRec dataset, where the experiment setting is same as the setting of other datasets. The corresponding results are given as follows:
>
> | Methods | AUC | Gain (AUC) |NDCG@50 | Gain (NDCG@50)|
> |:-:|:-:|:-:|:-:|:-:|
> |  naive |0.7498±0.0010 |- |0.7356±0.0012 |- |
> |  IPS |0.7314±0.0023 |- |0.7450±0.0015 |- |
> |  SNIPS |0.8015±0.0020 |- |0.8082±0.0007 |-|
> |  IPS-AT |0.7733±0.0063 |- |0.8003±0.0037 |-|
> |CVIB |0.7727±0.0064 |- |0.7852±0.0065 |- |
> |IPS-V2 |0.7787±0.0016 |- |0.7905±0.0029 |- |
> |D-IPS (Ours) |0.7947±0.0005 |8.65% |0.7876±0.0009 |5.71% |
> |D-SNIPS (Ours) |0.8026±0.0017 |0.137% |0.8084±0.0005 |0.247% |
> |D-IPS-AT (Ours) |0.7882±0.0042 |1.93% |0.8143±0.0023 |1.75% |
> |DR |0.7701±0.0058 |- |0.7818±0.0029 |- |
> |DR-JL |0.7808±0.0034 |- |0.7930±0.0033 |- |
> |MRDR-JL |0.7735±0.0008 |- |0.8121±0.0013 |- |
> |Stable DR |0.7812±0.0007 |- |0.7928±0.0040 |- |
> |Stable MRDR |0.7844±0.0013 |- |0.7752±0.0021 |- |
> |TDR-CL |0.7858±0.0016 |- |0.7776±0.0015 |- |
> |TMRDR-CL |0.7801±0.0017 |- |0.8047±0.0013 |- |
> |DR-V2 |0.7839±0.0027 |- |0.7923±0.0056 |- |
> |D-DR |0.7956±0.0064 |3.31% |0.7835±0.0040 |2.17% |
> |D-DR-JL |0.7742±0.0011 |-0.845% |0.7897±0.0017 |-0.416% |
> |D-MRDR-JL |0.7918±0.0011 |2.37% |0.8105±0.0010 |-0.197% |
>
> In the final version, we will merge this table into Table 2.
>
>
> A5. Thanks for your careful reading. We will correct the language issues and further improve the presentation in the final version.
>
> **Response for Questions:**
>
> A1. Sorry for the incorrect formula in line 757 and thanks very much for your careful reading. $'='$ should be $'\ge'$ in this formula.
>
> Denote $f(o _{u,i},\hat{p} _{u,i})e _{u,i}+g(o _{u,i},\hat{p} _{u,i})\hat{e} _{u,i}$ as $r(o _{u,i},\hat{p} _{u,i},e _{u,i},\hat{e} _{u,i})$. Then, the term in line 757 satisfies
> $$
> \begin{align}
> \frac{1}{\vert\mathcal{D}\vert^2}&\mathbb{E} _O\Bigg[\bigg(\sum _{(u,i)\in\mathcal{D}}r(o _{u,i},\hat{p} _{u,i},e _{u,i},\hat{e} _{u,i})\bigg)^2\Bigg]\\\\
> =&\frac{1}{\vert\mathcal{D}\vert^2}\sum _{j=1}^{\vert\mathcal{D}\vert}\sum _{k=1}^{\vert\mathcal{D}\vert}\mathbb{E} _O\Big[r(o _j,\hat{p} _j,e _j,\hat{e} _j)r(o _k,\hat{p} _k,e _k,\hat{e} _k)\Big]\\\\
> \ge&\frac{1}{\vert\mathcal{D}\vert^2}\sum _{j=1}^{\vert\mathcal{D}\vert}\mathbb{E} _O\big[r^2(o _j,\hat{p} _j,e _j,\hat{e} _j)\big]\quad\quad\quad\text{becase of } r(o _{u,i},\hat{p} _{u,i},e _{u,i},\hat{e} _{u,i})\ge0
> \end{align}$$
> Therefore, $\mathbb{V}_O[L]$ satisfies
> $$
> \begin{align}
> \mathbb{V} _O[L]\ge&\frac{1}{\vert\mathcal{D}\vert^2}\sum _{(u,i)\in\mathcal{D}}\frac{1-p _{u,i}}{p _{u,i}}\big[e _{u,i}-g(0,\hat{p} _{u,i})\hat{e} _{u,i}\big]^2.
> \end{align}
> $$
> Therefore, for any $e _{u,i}-g(0,\hat{p} _{u,i})\hat{e} _{u,i}\ne0$, $\lim _{p _{u,i}\to0}\mathbb{V} _O[L]=\infty$.
>
> We will use the above proof to replace the formulas in lines 757 and 758.
>
> A2. In the optimization problem (4), $E _B(\cdot)$ and $E _V(\cdot)$ are the measure metrics of bias and variance, respectively. As mentioned in **Bias-Variance Quantitative Joint Optimization**, (4) can be simplified as $w _1h _B^{Est}(\alpha _{u,i})+w _2h _V^{Est}(\alpha _{u,i})$. We can obtain the analytical solution of the optimal parameter $\alpha _{u,i}^{opt}$ given in Theorem 3.5.
>
> A3. Considering the principle **Isotonic Propensity**, the function $f(\hat{p} _{u,i})$ is a probability mapping. Therefore, $f(\hat{p} _{u,i})$ should be a monotonically increasing function with $f(0)=0$ and $f(1)=1$. $f(\hat{p} _{u,i})>\hat{p} _{u,i}$ guarantees $h _B^{Est}\ge0$ and $h _V^{Est}\ge0$. With this operation, (3) can be simplified as $\min _{\alpha _{u,i}}\{w _1h _B(\alpha _{u,i})+w _2h _V(\alpha _{u,i})\}$. Then, we can obtain the analytical solution of $\alpha _{u,i}^{opt}$ without increasing the computational complexity. Next, $\lim\limits _{\hat{p} _{u,i}\to0}\frac{\hat{p} _{u,i}}{f(\hat{p} _{u,i})}=C,\forall\alpha _{u,i}\in[0,1]$ ensures
> $$
> \begin{align}
> \lim _{\hat{p} _{u,i}\to0}h^{\text{Est}} _B(\hat{p} _{u,i},p _{u,i},\alpha _{u,i})=1-Cf^{1-\alpha _{u,i}}(0)
> \end{align}
> $$
> and
> $$
> \begin{align}
> \lim _{\hat{p} _{u,i}\to0,\alpha _{u,i}\leq0.5}h^{\text{Est}} _V(\hat{p} _{u,i},p _{u,i},\alpha _{u,i})=\lim _{\hat{p} _{u,i}\to0,\alpha _{u,i}\leq0.5}C(1-p _{u,i})f^{1-2\alpha _{u,i}}(\hat{p} _{u,i})=0
> \end{align}
> $$
> In final version, we will add the above discussion to explain these two function design principles.

---

> > ### Comment · Reviewer_rrSW · 2024-08-13
> >
> > Thank you for the response. I will maintain my positive score and increase confidence to 3.

---

### Official Review · Reviewer_aCF5 · 2024-07-03

**Soundness:** 3
**Presentation:** 3
**Contribution:** 3
**Rating:** 6
**Confidence:** 5

**Summary:**

This paper addresses the challenge of handling data missing-not-at-random, which is prevalent in applications like recommendation systems and display advertising. The authors highlight the limitations of current regularization techniques and unbiased estimators, which often result in high variance and unbounded generalization bounds. They propose a novel systematic fine-grained dynamic learning framework that jointly optimizes bias and variance. This framework adaptively selects the most appropriate estimator for each user-item pair according to a predefined objective function, ensuring reduced and bounded variances and generalization bounds with theoretical guarantees. Extensive experiments validate the theoretical findings and demonstrate the effectiveness of this dynamic learning approach in improving model stability and performance.

**Strengths:**

Originality
The paper introduces a novel approach to handling data missing-not-at-random (MNAR) with a fine-grained dynamic framework for bias-variance joint optimization. This method moves beyond traditional bias or variance reduction by addressing both simultaneously through dynamic estimator selection tailored to each user-item pair. This innovative dual optimization strategy significantly enhances the robustness and accuracy of predictive models in various applications.

Quality
The theoretical contributions are robust and well-grounded, with detailed mathematical derivations and proofs supporting the claims. The authors clearly identify the limitations of existing regularization techniques and unbiased estimators, providing a solid foundation for their proposed solution. Extensive experiments on real-world datasets validate the framework's effectiveness. Multiple performance metrics (AUC, NDCG) ensure a thorough assessment, demonstrating practical utility and reliability.

Clarity
The paper is well-structured and logically organized, making it relatively easy to follow despite the complexity. Each section builds on the previous one, creating a cohesive narrative from problem identification to theoretical underpinnings, proposed solution, and experimental validation. Clear headings, subheadings, mathematical formulations, and visual aids like tables and figures help in understanding key concepts and results. While some sections could be simplified, the overall presentation effectively communicates the core ideas.

Significance
The paper advances the state of the art in dealing with MNAR data, a common issue in many real-world applications like recommendation systems and online advertising. The framework dynamically optimizes bias and variance, addressing a critical gap in the literature and leading to more stable and accurate predictive models. Improved handling of MNAR data enhances the performance and reliability of machine learning systems across diverse domains. Theoretical insights into estimator design trade-offs can inspire future research and development.

**Weaknesses:**

One notable weakness is the experimental design. The experiments focus on AUC and NDCG metrics, which may not capture all aspects of model performance. Including metrics like precision, recall, or F1-score would offer a more comprehensive evaluation.

The paper also lacks a detailed comparison with state-of-the-art methods. While some comparisons are made, they are often superficial. The authors mention that their framework outperforms existing methods in bias and variance reduction but do not provide a detailed analysis of why this is the case. A thorough examination of the differences between their approach and other leading methods, focusing on specific scenarios, would strengthen the paper. A more rigorous ablation study to understand the contribution of each component of the proposed framework would also be beneficial.

The discussion on the practical implementation of the proposed framework is lacking. The authors provide little guidance on integrating their framework into existing systems or scaling it to handle large datasets. Practical considerations like computational complexity, memory requirements, and real-time applicability are not adequately addressed. More detailed information on these aspects, along with potential solutions or workarounds, would enhance the paper’s utility for practitioners.

Lastly, the paper could improve in presentation and readability. The writing is dense and technical, which may pose a barrier to understanding for readers unfamiliar with the subject matter. Breaking down complex ideas into simpler parts and including more visual aids like diagrams and flowcharts would make the paper more engaging and easier to follow. Clearer explanations and definitions of technical jargon would also help.

**Questions:**

1. In the experimental section, would other metrics such as precision, recall, or F1-score provide additional insights into the performance of your framework?
2. The paper mentions that the proposed framework dynamically selects the most appropriate estimator for each user-item pair. Could you provide more details on how these selections are made in practice, and whether there are computational complexities associated with this dynamic selection?
3. There is limited discussion on the practical implementation and scalability of your proposed framework. How does your approach handle large-scale datasets in terms of computational complexity and memory requirements?
4. The ablation study explores the contributions of different components of your framework. Could you provide more insights into the most critical components and how they individually contribute to the overall performance?
5. Your paper discusses the theoretical bounds for variance and generalization. Could you provide more practical insights or guidelines on how these bounds can be utilized or interpreted in real-world applications?
6. There are several functions for the dynamic estimators listed in Table 1. How were these functions selected, and could you discuss the potential impact of choosing different functions on the performance of your framework?
7. How sensitive is your method to the choice of hyperparameters, and what guidelines would you provide for selecting these parameters in practical applications?

**Limitations:**

The authors have made a commendable effort in addressing their work's limitations by thoroughly discussing the theoretical challenges and constraints of existing regularization techniques. However, additional detail would enhance the paper’s comprehensiveness, especially regarding the practical limitations of the proposed framework in terms of computational complexity and scalability. Understanding how the framework performs with large-scale datasets and the resources required for its implementation would benefit practitioners.

Regarding the broader societal impact, the authors briefly mention potential applications but do not deeply explore societal implications. It would be valuable to discuss how their framework might affect users' privacy, data security, and potential biases in recommendation systems. Addressing potential negative consequences and proposing mitigation strategies would show a thorough consideration of ethical concerns and enhance the paper’s robustness.

To improve, the authors could include a dedicated section on the societal impacts of their work, examining both positive and negative aspects. This section should address how the framework could be used responsibly, ensuring it benefits society while minimizing potential harms. Additionally, incorporating feedback from stakeholders or conducting a preliminary ethical review could provide further insights and strengthen this discussion.

---

> ### Author Rebuttal · Authors · 2024-08-07
>
> A1. Thanks for your valuable comments. In the final version, we will add more metrics. Besides, the baseline approaches and the dynamic estimators are conducted on the KuaiRec dataset to verify the performance of the developed dynamic fine-grained framework. The experimental results will be merged into Table 1.
>
> A2. Based on $\alpha _{u,i}\in[0,1]$, when $w _2/w _1$ satisfies $\frac{w _2}{w _1}\leq\frac{f(p _{u,i})}{2(1-p _{u,i})}$, $\alpha _{u,i}^{opt}=1$; when  $\frac{f(p _{u,i})}{2(1-p _{u,i})}\leq\frac{w _2}{w _1}\leq\frac{1}{2(1-p _{u,i})}$, then $\alpha _{u,i}^{opt}=\frac{\ln\Big(\frac{2w _2}{w _1}(1-p _{u,i})\Big)}{\ln(f(p _{u,i}))}$; when $\frac{w _2}{w _1}\ge\frac{1}{2(1-p _{u,i})}$, then $\alpha _{u,i}^{opt}=0$. Therefore, the daynamic estimator can be rewritten as
> $$\begin{align}L _{D-IPS}=&\frac{1}{\vert\mathcal{D}\vert}\sum _{(u,i)\in\mathcal{D}}\mathbb{1} _{\bigg[\frac{w _2}{w _1}\leq\frac{f(p _{u,i})}{2(1-p _{u,i})}\bigg]}\frac{o _{u,i}}{f(\hat{p} _{u,i})}e _{u,i}+\mathbb{1} _{\bigg[\frac{f(p _{u,i})}{2(1-p _{u,i})}\leq\frac{w _2}{w _1}\leq\frac{1}{2(1-p _{u,i})}\bigg]}\frac{o _{u,i}}{f^{\alpha^{opt} _{u,i}}(\hat{p} _{u,i})}e _{u,i}
> +\mathbb{1} _{\bigg[\frac{w _2}{w _1}\ge\frac{1}{2(1-p _{u,i})}\bigg]}o _{u,i}e _{u,i},\\\\
> L _{D-DR}=&\frac{1}{\vert\mathcal{D}\vert}\sum _{(u,i)\in\mathcal{D}}\mathbb{1} _{\bigg[\frac{w _2}{w _1}\leq\frac{f(p _{u,i})}{2(1-p _{u,i})}\bigg]}\bigg(\hat{e} _{u,i}+\frac{o _{u,i}}{f(\hat{p} _{u,i})}\delta _{u,i}\bigg)
> +\mathbb{1} _{\bigg[\frac{f(p _{u,i})}{2(1-p _{u,i})}\leq\frac{w _2}{w _1}\leq\frac{1}{2(1-p _{u,i})}\bigg]}\bigg(\hat{e} _{u,i}+\frac{o _{u,i}}{f^{\alpha^{opt} _{u,i}}(\hat{p} _{u,i})}\delta _{u,i}\bigg)
> +\mathbb{1} _{\bigg[\frac{w _2}{w _1}\ge\frac{1}{2(1-p _{u,i})}\bigg]}\bigg(\hat{e} _{u,i}+o _{u,i}\delta _{u,i}\bigg).
> \end{align}$$
> Therefore, the proposed framework dynamically selects the most appropriate estimator for each user-item pair.
>
> $f(\hat{p} _{u,i})>\hat{p} _{u,i}$ guarantees $h _B^{Est}\ge0$ and $h _V^{Est}\ge0$. With this operation, the joint optimization problem (3) can be simplified as $\min _{\alpha _{u,i}}\{w _1h _B(\alpha _{u,i})+w _2h _V(\alpha _{u,i})\}$. Then, we can obtain the analytical solution of $\alpha _{u,i}^{opt}$ without increasing the computational complexity.
>
> A3. In (5), we have given the analytical solution of the optimal parameter. If an engineer have an IPS-based or DR-based estimators for large-scale datasets, then she/he can modify the IPS-based or DR-based estimators into a D-IPS-based or D-DR-based estimator by the following core code.
> ```
> w1 = 1
> w2 = 0.5
> star = 2 * w2 * (1 - propensity) / w1
> alpha = np.log(star) / np.log(propensity)
> lower_bound = np.zeros(np.size(propensity))
> upper_bound = np.ones(np.size(propensity))
> alpha = np.where(alpha > upper_bound, upper_bound, alpha)
> alpha = np.where(alpha < lower_bound, lower_bound, alpha)
> propensity = np.power(propensity, alpha)
> ```
>
> A4. In the proposed dynamic framework, the weight ratios and the function forms jointly determined the performance of estimators. As mentioned in A2, when $\frac{w _2}{w _1}=0$ and $f(\hat{p} _{u,i})=\hat{p} _{u,i}$, then the optimal parameter $\alpha _{u,i}^{opt}$ is set as 1, and the D-IPS and D-DR methods are equivalent to IPS and DR methods, which are unbiased approaches. According to Fig. 2, we can obeserved the performance of dynamic estimators under different weight ratio, which includes the case of $\frac{w _2}{w _1}=0$. On the other hand, under the identical weight ratio $\frac{w _2}{w _1}=0.1$, effects of different functions on performances of dynamic estimators are shown in Table 3. From $Gain _{AUC}$ and $Gain _{N}$ in Table 3, we can observe the individual contribution of different function forms.
>
> A5. The definition of the variance reveals the preformance stability of the estimator in whole propensity space. For a new dataset, we cannot know in advance the range of the propensities in this dataset. An unknown biased dataset may disrupt the stability of the prediction model, which brings significant risks to practical applications. Therefore, it is necessary to consider the boundedness of variance and generalization to ensure the stability and performance of prediction models. In real-world applications, the theoretical bounds for variance and generalization indrectly reflect the stability and generalization performance of estimators, respectively.
>
> A6. We find that the function $f(\hat{p} _{u,i})$ is not unique. According to the functon design principles, we can obtain a family of functions. According to Fig. (c), for a fixed propensity, different functions correspond to different minimum objective values. For different datasets with different propensity distributions, we can determine the function form according to the minimum objective values of functions under the propensity distribution. Actually, there is further optimization potential for the selection of the function form. We can incorporate the selection of function forms into the bias-variance joint optimization problem (4). The corresponding optimization problem can be rewritten as
> $${Objective}^{opt}=\min _{\alpha _{u,i}\in[0,1], f\in\mathcal{F}}\Big[w _1E _B(h^{Est} _B(f^{\alpha _{u,i}}(\hat{p} _{u,i})))+w _2E _V(h^{Est} _V(f^{\alpha _{u,i}}(\hat{p} _{u,i})))\Big],{s.t.} 0\leq\alpha _{u,i}\leq1,$$
> where $\mathcal{F}$ is the candidate function set. This problem is one of our future works.
>
> A7. $w _1$ and $w _2$ in Eq. (3) are the hyper-parameter. $\alpha _{u,i}^{opt}$ only depends on $w _2/w _1$. $w _2/w _1$ will leads to the Pareto frontier of the estimation performance. As of now, optimization problem for weight ratios is still an open question because of unknown properties of its Pareto frontier. In practice, for a new dataset, the tuning range of $w _2/w _1$ is usually set to be between 0 and 1. Then several $w _2/w _1$ are used to test the estimation performance and the optimal weight ratio is selected as the final parameter.

---

> > ### Comment · Reviewer_aCF5 · 2024-08-13
> >
> > Thank you to the authors for their detailed rebuttal, which has addressed most of my concerns. I will maintain my positive rating.

---

### Official Review · Reviewer_h92D · 2024-07-13

**Soundness:** 3
**Presentation:** 3
**Contribution:** 3
**Rating:** 7
**Confidence:** 4

**Summary:**

In recommender systems, there are many ratings that are missing not at random, which leads to additional bias in RS when trained using only observed data. This paper first gives a general form of the estimator with regularization, then reveals limitations of previous regularization techniques and the relationship between unbiasedness of the generalized estimator and its generalization bound. In addition, this paper finds an interesting phenomenon that Regularization $L_{Reg}$ cannot guarantee a bounded variance and generalization bound for previous estimators. Then this paper proposes a bias-variance quantitative joint optimization approach with theoretical guarantees that the proposed estimator has a bounded variance and generalization bound. Extensive experiments on two real-world datasets validate the effectiveness of the proposed methods.

**Strengths:**

S1: Debias is a very important topic in the recommendation field. This paper is well-written with clear organization.

S2: This paper summarizes the general form of previous estimators, and then proposes an dynamic estimators with sound theoretical results.

S3: This paper provides a novel bias-variance quantitative joint optimization algorithm that can make the proposed estimator has bounded variance and generalization bound.

S4: The experiments are sufficient and comprehensive. The results validate the effectiveness of the proposed methods. In addition, the experiments are conducted on public datasets which makes it easier to follow.

**Weaknesses:**

W1: The detailed proof of why Cov($L_{Est}$, $L_{Reg}$) $\geq 0$ when $E_O[L_{Reg}] = 0$ is needed. In addition, the proof for Corollary
3.2 is needed.

W2: How to determine the explicit form of function $f^{\alpha}(\hat p_{u,i})$ in practice?

W3: Authors should provide the detailed learning algorithm step-by-step to explain how the propensity model, prediction model and imputation model are learned.

W4: Missing some recent references. There are many works focusing on the debiased recommendation. For example, [1] proposes a kernel balancing methods for learning propensity model, [2] proposes a doubly calibrated estimator that involves the calibration of both
the imputation and propensity models, and [3] proposes a multiple robust estimator combining multiple propensity and imputation models.

W5: Some Typos. For example, it should be Appendix C in line 144 instead of Appendix D and some unexpected indents after the formula like line 203 should be removed.

[1] Debiased Collaborative Filtering with Kernel-Based Causal Balancing. In ICLR, 2024.

[2] Doubly Calibrated Estimator for Recommendation on Data Missing Not At Random. In WWW, 2024

[3] Multiple Robust Learning for Recommendation. In AAAI, 2023

**Questions:**

See the Weaknesses part above for the questions.

**Limitations:**

Yes, the authors adequately discussed and addressed the limitations of this paper.

---

> ### Author Rebuttal · Authors · 2024-08-06
>
> A1. Considering (1), for all $(u,i)$ pairs, we have $f(o _{u,i},\hat{p} _{u,i})e _{u,i}+g(o _{u,i},\hat{p} _{u,i})\hat{e} _{u,i}\ge0$ and $h(o _{u,i},\hat{p} _{u,i})$. To facilitate representation, $f(o _{u,i},\hat{p} _{u,i})e _{u,i}+g(o _{u,i},\hat{p} _{u,i})\hat{e} _{u,i}$ is denoted as $r(o _{u,i},\hat{p} _{u,i},e _{u,i},\hat{e} _{u,i})$. In the proof of Theorem 3.1, $\text{Cov}(L _\text{Est}, L _\text{Reg})$ satisfies
> $$\begin{align}\text{Cov}(L _\text{Est}, L _\text{Reg})=&\frac{1}{\vert\mathcal{D}\vert^2}\mathbb{E} _O\Bigg(\sum _{(u,i)\in\mathcal{D}}\bigg[h(o _{u,i},\hat{p} _{u,i})\sum _{(u,i)\in\mathcal{D}}r(o _{u,i},\hat{p} _{u,i},e _{u,i},\hat{e} _{u,i})\bigg]\Bigg)\nonumber\\\\=&\frac{1}{\vert\mathcal{D}\vert^2}\mathbb{E} _O\Bigg(\sum _{j=1}^{\vert{D}\vert}\sum _{k=1}^{\vert{D}\vert}h(o _j,\hat{p} _j)r(o _k,\hat{p} _k,e _k,\hat{e} _k)\Bigg)\nonumber\\\\=&\frac{1}{\vert\mathcal{D}\vert^2}\sum _{j=1}^{\vert{D}\vert}\sum _{k=1}^{\vert{D}\vert}\mathbb{E} _O[h(o _j,\hat{p} _j)r(o _k,\hat{p} _k,e _k,\hat{e} _k)].\end{align}$$
> Since $\mathbb{E} _O[h(o _j,\hat{p} _j)r(o _k,\hat{p} _k,e _k,\hat{e} _k)]\ge0$, we obtain $\text{Cov}(L _{Est}, L _{Reg})\ge0$.
>
> We will revise this sentense in line 124 as follows, and add the above formula (2) to proof of Theorem 3.1.
>
> *"... respectively. For all $(u,i)$ pairs, they satisfy $f(o _{u,i},\hat{p} _{u,i})e _{u,i}+g(o _{u,i},\hat{p} _{u,i})\hat{e} _{u,i}\ge0$ and $h(o _{u,i},\hat{p} _{u,i})$. $\lambda>0$ is a ..."*
>
> Since Corollary 3.2 is the contrapositive of Theorem 3.1, the proof of Theorem 3.1 has been given. Therefore, Corollary 3.2 also holds. We further provide the proof of Corollary 3.2 as follows
>
> **proof of Corollary 3.2**
>
> We use the method of proof by contradiction. Assume that when $\mathbb{V} _O[L _\text{Est+Reg}]\leq\mathbb{V} _O[L _\text{Est}]$, $L _\text{Est+Reg}$ is unbiased.
>
> According to $\mathbb{V} _O[L _\text{Est+Reg}]\leq\mathbb{V} _O[L _\text{Est}]$, we have
> $$\begin{align}
> \mathbb{V} _O[L _\text{Est+Reg}]=&\mathbb{V} _O[L _\text{Est}]+2\lambda\text{Cov}(L _\text{Est}, L _\text{Reg})+\lambda^2\mathbb{V} _O[L _\text{Reg}]\nonumber\\\\
> \leq&\mathbb{V} _O[L _\text{Est}],
> \end{align}$$
> which implies that $2\lambda\text{Cov}(L _\text{Est}, L _\text{Reg})+\lambda^2\mathbb{V} _O[L _\text{Reg}]\leq0$. Therefore, the parameter $\lambda$ needs to satisfy $0\leq\lambda\leq-\frac{2\text{Cov}(L _\text{Est}, L _\text{Reg})}{\mathbb{V} _O[L _\text{Reg}]}$, which implies $\text{Cov}(L _\text{Est}, L _\text{Reg})\leq0$.
>
> As shown in A1, when $L _\text{Est+Reg}$ is unbiased, we have $\text{Cov}(L _{Est}, L _{Reg})\ge0$, which contradicts the condition $\text{Cov}(L _\text{Est}, L _\text{Reg})\leq0$. Therefore, Corollary 3.2 holds.
>
> A2. We can obtain a family of functions. According to $w _1h _B^{Est}(\alpha,p _{u,i})+w _2h _V^{Est}(\alpha,p _{u,i})$ and Fig. (c), we can observe that different $f^{\alpha _{u,i}}(\hat{p} _{u,i})$ will lead to different objective function surface. For different datasets with different propensity distributions, we can determine the function form according to the minimum objective values of functions. Actually, there is further optimization potential for the selection of the function form. We can incorporate the selection of function forms into the bias-variance joint optimization problem (4), which is formulated as
> $${Objective}^{opt}=\min _{\alpha _{u,i}\in[0,1], f\in\mathcal{F}}\Big[w _1E _B(h^{Est} _B(f^{\alpha _{u,i}}(\hat{p} _{u,i})))+w _2E _V(h^{Est} _V(f^{\alpha _{u,i}}(\hat{p} _{u,i})))\Big],{s.t.} 0\leq\alpha _{u,i}\leq1$$
> This problem is one of our future works.
>
> A3. This work focus on revealing the general rules of regularization techniques and unbiased estimators and developed a fine-grained dynamic framework to jointly optimize bias and variance. Therefore, almost all learning algorithms can be transformed into the fine-grained dynamic framework dynamic framework, such as joint learning of DR, double learning of MRDR, cycle learning of SDR, collaborative learning of TDR. In the final version, we will add the following core code of fine-grained dynamic estimators to Appendix E to explain how to transform a propensity-based debiased approach to the fine-grained dynamic estimator.
>
> **Core Code of Fine-Grained Dynamic Estimators**
>
> ```
> w1 = 1
> w2 = 0.5
> rate = w2 / w1
> star = 2 * rate * (1 - propensity)
> # corresponds to function 1 given in Table 1
> alpha = np.log(star) / np.log(propensity)
> # alpha = np.log(star) / np.log(np.sin(propensity)/np.sin(1))
> # alpha = np.log(star) / np.log(np.log(propensity+1)/np.log(2))
> # alpha = np.log(star) / np.log(np.tanh(propensity)/np.tanh(1))
> lower_bound = np.zeros(np.size(propensity))
> upper_bound = np.ones(np.size(propensity))
> alpha = np.where(alpha > upper_bound, upper_bound, alpha)
> alpha = np.where(alpha < lower_bound, lower_bound, alpha)
> propensity = np.power(propensity, alpha)
> ```
> A4. In the final version, we will add the following descriptions in line 313 to improve Related Work.
>
> *“...forth. To improve the robustness of estimators, a multiple robust estimator is developed in [3] by taking the advantage of multiple candidate imputation and propensity models, which is unbiased when any of the imputation or propensity models, or a linear combination of these models is accurate. From a novel function balancing perspective, Li et al. propose to approximate the balancing functions in reproducing kernel Hilbert space and design adaptively kernel balancing IPS/DR learning algorithms [1]. Moreover, aimed at limitations of miscalibrated imputation and propensity models, Kweon and Yu [2] propose a doubly calibrated estimator and a tri-level joint learning framework to simultaneously optimize calibration experts alongside prediction and imputation models. For the...”*
>
> A5. Thanks for your careful reading. We will correct typos, language issues, and unclear desciption, and further improve the presentation in the final version.

---

> > ### Comment · Reviewer_h92D · 2024-08-13
> >
> > I read the authors rebuttal very carefully. Well done for the authors! I thank the authors for their great efforts and suggest that these revisions be incorporated in their final version. I'm happy to raise my score to 7.

---

### Official Review · Reviewer_AciS · 2024-07-17

**Soundness:** 2
**Presentation:** 3
**Contribution:** 3
**Rating:** 5
**Confidence:** 4

**Summary:**

This paper first theoretically reveals the limitations of previous regularization techniques, such as unbounded variance and generalization bound. To address this problem, this paper defines the general form of the estimator with regularization. Then this paper develops a comprehensive dynamic learning framework, which can lead to reduced and bounded generalization bounds and variances. Experiments on two real-world datasets verify the theoretical results and the performance of the proposed method.

**Strengths:**

$\bullet~$ The problem studied is important and relevant.

$\bullet~$ The idea is interesting and novel.

$\bullet~$ The theoretical results are sound.

$\bullet~$ The evaluations are solid and convincing.

**Weaknesses:**

$\bullet~$ The analysis of Figure 1 is missing.

$\bullet~$ There is no $\alpha_{u, i}$ in $lim$ in line 189. In addition, is the constant $C$ related to $\alpha_{u, i}$?

$\bullet~$ Does the $w_1$ and $w_2$ in equation 3 be the hyper-parameter? How can we determine such hyper-parameters in practice and what is the tuning range for the experiments?

$\bullet~$ What is the unbiased condition for the proposed methods? Meanwhile, what is the bias when the optimal In addition, experiments according to the data sparsity should be conducted. If the observed data is not sparse, there may be no propensity score that will not tend to 0.

$\bullet~$ Some typos. For example, it should be $\mathbb E_O[L^2_{Est}] + \lambda^2 \mathbb E_O[L^2_{Reg}] + 2 \lambda \mathbb E_O[L_{Est} L_{Reg}]$ in equation 10. Line 142 "Theorems" should be "Theorem".

**Questions:**

I have some questions about the proofs: \
First, for the proof of Theorem 3.1, why Cov($L_{Est}$, $L_{Reg}$) $\geq 0$? \
Second, the description of Theorem 3.1 is vague. What is the meaning of "greater than the one of the original estimator"? In addition, in the current version, the contrapositive of Theorem 3.1 is "less than any of the original estimators", instead of "less than one of the original estimator" in Corollary 3.2. \
Third, for the proof of Theorem 3.3, I understand the $\mathbb E_O[L^2_{Est}]$ tends to infinity, but why $\mathbb E_O[L^2_{Est}] + \lambda \mathbb E_O[L^2_{Reg}] + 2 \lambda^2 \mathbb E_O[L_{Est} L_{Reg}]$ tends to infinity? In the extreme case, let $L_{Reg}  = -L_{Est}$ and $\lambda = 1$, then $\mathbb V_O[L_{Est+Reg}] = 0$.

**Limitations:**

Yes

---

> ### Author Rebuttal · Authors · 2024-08-06
>
> **Response for Weaknesses:**
> -  In the final version, we will add the following analysis of Figure 1 to lines 212 and 220, respectively.
>
> *''**Line 212:**.... The curves of objective functions under different designed functions $f(\cdot)$ are given in Fig. 1(c). It can be oberved that for a fixed propensity, there exists an $\alpha$ such that the objective function attains the minimum value. Besides, ...''*
>
> *''**Line 220:**.... Under different designed function $f(\cdot)$, the schematic diagram of optimal objective values corresponding to the optimal parameter $\alpha_{u,i}^{opt}$ is shown in Figure 1(d). Next, ...''*
>
> - The constant $C$ is not related to $\alpha_{u,i}$. In line 189, what we want to convey is that for all $\alpha_{u,i}\in[0,1]$, the constant $C$ is the same.
>
> - Yes, $w_1$ and $w_2$ in Eq. (3) are the hyper-parameter. As shown in (5), $\alpha_{u,i}^{opt}$ depends on the weight ratio $w_2/w_1$. The weight ratio space will leads to the Pareto frontier of prediction model performance. As disussed in Conclusions, various properties of the Pareto frontier and optimization methods for weight ratios are still an open question, and it is one of our future works. In practice, the tuning range of $w_2/w_1$ is usually set to be between 0 and 1.
>
> - In Theorem 3.5, when $\frac{w_2}{w_1}=0$ and $f(\hat p_{u,i})=\hat p_{u,i}$, then the optimal parameter $\alpha_{u,i}^{opt}$ is set as 1, and D-IPS and D-DR are equivalent to IPS and DR, which are unbiased. Therefore, the unbiased conditions are $\frac{w_2}{w_1}=0$, $f(\hat p_{u,i})=\hat p_{u,i}$ when propensities or imputation errors are accurate. When $\alpha$ is optimal, according to the Lemma D.1, the bias of the proposed estimator can be obtained. When $\frac{w_2}{w_1}\leq\frac{f(p_{u,i})}{2(1-p_{u,i})}$, then $\alpha_{u,i}^{opt}=1$; when  $\frac{f(p_{u,i})}{2(1-p_{u,i})}\leq\frac{w_2}{w_1}\leq\frac{1}{2(1-p_{u,i})}$, then $\alpha_{u,i}^{opt}=\frac{\ln\Big(\frac{2w_2}{w_1}(1-p_{u,i})\Big)}{\ln(f(p_{u,i}))}$; when  $\frac{w_2}{w_1}\ge\frac{1}{2(1-p_{u,i})}$, then $\alpha_{u,i}^{opt}=0$. Therefore, the dynamic estimators are applicable to different propensity distributions by adjusting the weight ratio $w _2/w _1$. Different weight ratios will lead to different distributions of instances on different estimators. Therefore, even if there is no propensity score that will not tend to 0, the developed fine-grained dynamic framework still can achieve quantitative optimization of bias and variance.
>
> - Equation (10) should be
> $$\begin{align}
> \mathbb{V} _O[L _\text{Est+Reg}]
> =\mathbb{E} _O[L^2 _\text{Est}+\lambda^2{L}^2 _\text{Reg}+2\lambda{L} _\text{Est}L _\text{Reg}]-\mathbb{E}^2 _O[L _\text{Est}+\lambda{L} _\text{Reg}].
> \end{align}$$
>
> **Response for Questions:**
>
> A1. Considering (1), for all $(u,i)$ pairs, we have $f(o _{u,i},\hat{p} _{u,i})e _{u,i}+g(o _{u,i},\hat{p} _{u,i})\hat{e} _{u,i}\ge0$ and $h(o _{u,i},\hat{p} _{u,i})\ge0$. To facilitate representation, $f(o _{u,i},\hat{p} _{u,i})e _{u,i}+g(o _{u,i},\hat{p} _{u,i})\hat{e} _{u,i}$ is denoted as $r(o _{u,i},\hat{p} _{u,i},e _{u,i},\hat{e} _{u,i})$.  As given in the proof of Theorem 3.1, $\text{Cov}(L _\text{Est}, L _\text{Reg})$ satisfies
> $$\begin{align}\text{Cov}(L _\text{Est}, L _\text{Reg})=&\frac{1}{\vert\mathcal{D}\vert^2}\mathbb{E} _O\Bigg(\bigg[\sum _{(u,i)\in\mathcal{D}}r(o _{u,i},\hat{p} _{u,i},e _{u,i},\hat{e} _{u,i})\bigg]\bigg[\sum _{(u,i)\in\mathcal{D}}h(o _{u,i},\hat{p} _{u,i})\bigg]\Bigg)\nonumber\\\\=&\frac{1}{\vert\mathcal{D}\vert^2}\mathbb{E} _O\Bigg(\sum _{(u,i)\in\mathcal{D}}\bigg[h(o _{u,i},\hat{p} _{u,i})\sum _{(u,i)\in\mathcal{D}}r(o _{u,i},\hat{p} _{u,i},e _{u,i},\hat{e} _{u,i})\bigg]\Bigg)\nonumber\\\\=&\frac{1}{\vert\mathcal{D}\vert^2}\mathbb{E} _O\Bigg(\sum _{j=1}^{\vert{D}\vert}\sum _{k=1}^{\vert{D}\vert}h(o _j,\hat{p} _j)r(o _k,\hat{p} _k,e _k,\hat{e} _k)\Bigg)\nonumber\\\\=&\frac{1}{\vert\mathcal{D}\vert^2}\sum _{j=1}^{\vert{D}\vert}\sum _{k=1}^{\vert{D}\vert}\mathbb{E} _O[h(o _j,\hat{p} _j)r(o _k,\hat{p} _k,e _k,\hat{e} _k)].\end{align}$$
> Since $\mathbb{E} _O[h(o _j,\hat{p} _j)r(o _k,\hat{p} _k,e _k,\hat{e} _k)]\ge0$, we obtain $\text{Cov}(L _{Est}, L _{Reg})=\mathbb{E} _O(L _\text{Est}L _\text{Reg})\ge0$.
>
> We will revise this sentense in line 124 as follows, and add the above formula (1) to proof of Theorem 3.1.
>
> *... respectively. For all $(u,i)$ pairs, they satisfy $f(o _{u,i},\hat{p} _{u,i})e _{u,i}+g(o _{u,i},\hat{p} _{u,i})\hat{e} _{u,i}\ge0$ and $h(o _{u,i},\hat{p} _{u,i})\ge0$. $\lambda>0$ is a ...*
>
> A2. 'the one' in Theorem 3.1 and Corollary 3.2 is a pronoun that refers to the variance. What we want to convey in Theorem 3.1 and Corollary 3.2 are that
>
> *"**Theorem 3.1** Let $L _\text{Est+Reg}$ be defined in (1) and the estimator $L _\text{Est}$ be unbiased. If $L _\text{Est+Reg}$ is unbiased, then the variance of $L _\text{Est+Reg}$ is greater than the variance of the original estimator $L _\text{Est}$."*
>
> *"**Corollary 3.2** If the variance of $L _\text{Est+Reg}$ is less than the variance of the original estimator $L _\text{Est}$, then $L _\text{Est+Reg}$ is not unbiased."*
>
> In the final version, we will modify Theorem 3.1 and Corollary 3.2 to the descriptions above to avoid any ambiguity.
>
> A3. According to the proof of Theorem 3.3, we can obtain
> $$\begin{align}\mathbb{V} _O[L _\text{Est+Reg}]\ge\mathbb{E} _O[L^2 _\text{Est}]+\lambda^2\mathbb{E} _O[{L}^2 _\text{Reg}]+2\lambda\mathbb{E} _O[L _\text{Est}L _\text{Reg}]-\bar{B}^2.\nonumber\end{align}$$
>
> Based on  A1., we can obtain $\lambda^2\mathbb{E} _O[{L}^2 _\text{Reg}]\ge0$ and $\text{Cov}(L _{Est}, L _{Reg})\ge0$. Since $\mathbb{E} _O[L^2 _\text{Est}]$ tends to infinity, $\mathbb{E} _O[L^2 _\text{Est}]+\lambda^2\mathbb{E} _O[{L}^2 _\text{Reg}]+2\lambda\mathbb{E} _O[L _\text{Est}L _\text{Reg}]$ also tends infinity. For all estimators, ${L} _\text{Est}\ge0$ and ${L} _\text{Reg}\ge0$, ${L} _\text{Est}=-{L} _\text{Reg}$ is true only when ${L} _\text{Est}={L} _\text{Reg}=0$.

---

> > ### Comment · Reviewer_AciS · 2024-08-13
> >
> > Thanks for the rebuttal, which addresses my concerns. I will keep my rating unchanged.

---

### Decision · Program_Chairs · 2024-09-25

**Decision:**

Accept (poster)

**Comment:**

This paper highlights the limitations of existing regularization techniques in debiased recommendations and introduces a general bias-variance joint optimization approach to overcome these challenges. The reviewers found the authors' responses to be very helpful in addressing their concerns, with one reviewer even upgrading their score. Overall, all three reviewers provided positive evaluations.